# Reconfigurations of cortical manifold structure during reward-based motor learning

Qasem Nick[1,2], Daniel J Gale[1], Corson Areshenkoff[1,2], Anouk De Brouwer[1], Joseph Nashed[1,3], Jeffrey Wammes[1,2], Tianyao Zhu[1], Randy Flanagan[1,2], Jonny Smallwood[1,2], Jason Gallivan[1,2,4]*

[1]Centre for Neuroscience Studies, Queen's University, Kingston, Canada; [2]Department of Psychology, Queen's University, Kingston, Canada; [3]Department of Medicine, Queen's University, Kingston, Canada; [4]Department of Biomedical and Molecular Sciences, Queen's University, Kingston, Canada

*For correspondence:
gallivan@queensu.ca

**Abstract** Adaptive motor behavior depends on the coordinated activity of multiple neural systems distributed across the brain. While the role of sensorimotor cortex in motor learning has been well established, how higher-order brain systems interact with sensorimotor cortex to guide learning is less well understood. Using functional MRI, we examined human brain activity during a reward-based motor task where subjects learned to shape their hand trajectories through reinforcement feedback. We projected patterns of cortical and striatal functional connectivity onto a low-dimensional manifold space and examined how regions expanded and contracted along the manifold during learning. During early learning, we found that several sensorimotor areas in the dorsal attention network exhibited increased covariance with areas of the salience/ventral attention network and reduced covariance with areas of the default mode network (DMN). During late learning, these effects reversed, with sensorimotor areas now exhibiting increased covariance with DMN areas. However, areas in posteromedial cortex showed the opposite pattern across learning phases, with its connectivity suggesting a role in coordinating activity across different networks over time. Our results establish the neural changes that support reward-based motor learning and identify distinct transitions in the functional coupling of sensorimotor to transmodal cortex when adapting behavior.

## eLife assessment

This **valuable** study uses **convincing** state-of-the-art neuroimaging analyses to characterize whole-brain networks during reward-based motor learning. This work motivates future research to dissociate why the observed changes in neural connectivity occur and how they support reward-based motor learning. The study is highly relevant for researchers at the intersection of decision-making and sensorimotor learning.

## Introduction

Organizing our behaviors so that they match the demands of a given situation depends on establishing contingencies between specific features of an action and whether they lead to the desired outcome. In many real-world tasks, this is a challenging endeavor as the brain must learn how to modify its actions based on a single measure of performance feedback that reflects overall task success (*Berniker and Kording, 2008*; *Dhawale et al., 2017*; *Houk et al., 1996*; *Wolpert et al., 2001*).

Moreover, this feedback must be communicated to multiple discrete neural systems distributed across the cortex and striatum, many of which are topographically segregated from one another (*Averbeck and O'Doherty, 2022*). Numerous studies have shown that neural systems anchored within the medial prefrontal cortex (MPFC) and striatum are important in evaluating whether the results of behavior are in line with expectations (*Averbeck and O'Doherty, 2022*; *Klein-Flügge et al., 2022*; *Lee et al., 2012*; *O'Doherty et al., 2017*). Specifically, when discrepancies arise between the expected versus actual results of an action—termed a 'prediction error'—this information serves as the teaching signal that can be used to update behavior directly (*Bayer and Glimcher, 2005*; *O'Doherty et al., 2003*; *Samejima et al., 2005*; *Schultz et al., 1997*). How exactly this information is communicated in a coherent manner across the multiple, distributed neural systems that guide behavior remains poorly understood. Our study addresses this gap in our knowledge using state-of-the-art manifold learning techniques to describe how the landscape of brain activity changes during reward-guided motor learning.

Contemporary systems and cognitive neuroscience have identified many large-scale neural systems that have each been linked to different components of effective behavior. For example, areas in sensory cortex provide representations of the external environment, whereas areas in motor cortex are involved in generating the final motor commands required for action. At higher levels in the cortical hierarchy, regions within the frontoparietal system, along with those that make up the brain's attention-orienting systems (dorsal and ventral attention systems), are important in the selection of sensory inputs and the guidance of rule-driven behavior (*Corbetta et al., 2008*; *Corbetta and Shulman, 2002*). Yet, how the activity of these various brain systems is coordinated during the learning process is unclear. An emerging literature suggests that this coordination may depend, in part, on functional activity in several key regions of higher-order association cortex, known collectively as the default mode network (DMN) (*Buckner et al., 2008*; *Fox and Raichle, 2007*; *Margulies et al., 2016*; *Raichle, 2015*; *Smallwood et al., 2021*).

Initially identified through its tendency to deactivate during cognitively demanding tasks, the DMN has traditionally been implicated in largely introspective, abstract cognitive functions such as autobiographical memory and internal mentation (*Andrews-Hanna et al., 2014*; *Binder et al., 2009*; *Christoff et al., 2016*; *Schacter et al., 2012*; *Spreng et al., 2009*). In recent years, however, this characterization of DMN activity has been difficult to reconcile, with an emerging body of functional MRI (fMRI) and neurophysiological evidence showing that areas of this network are activated during demanding decision-making and working-memory tasks (*Foster et al., 2023*; *Hayden et al., 2009*; *Heilbronner and Platt, 2013*; *Murphy et al., 2019*; *Murphy et al., 2018*; *Pearson et al., 2009*; *Vatansever et al., 2017*). One hypothesis concerning this system's function pertains to its unique topographic positioning on the cortical mantle (*Smallwood et al., 2021*): Each core region of the DMN is located in regions of association cortex that are equidistant between different primary systems; for example, posteromedial cortex, a key node of the DMN, is located precisely at the midpoint between the calcarine (visual) and central (motor) sulci (*Margulies et al., 2016*). This unique topographic location is hypothesized to allow DMN regions broad oversight over distributed brain functions, enabling them to play a role in the coordination of activity across cortex (*Smallwood et al., 2021*).

Consistent with this contemporary perspective, recent work implicates several regions of the DMN in organizing different modes of behavior over time. For instance, DMN areas such as medial frontal cortex and posteromedial cortex appear to play an important role in shifting between information gathering versus information exploitation during reward-guided decision-making tasks (*Barack et al., 2017*; *Foster et al., 2023*; *Pearson et al., 2011*; *Pearson et al., 2009*; *Schuck et al., 2015*; *Trudel et al., 2021*)—the so-called explore/exploit trade-off (*Frank et al., 2009*; *Sutton and Barto, 2018*). Consistent with this, recent studies have argued that broad features of the DMN's activity can be explained under the auspices that it supports behavior under conditions in which performance depends on knowledge accrued across several trials rather than by immediate sensory inputs (*Hayden et al., 2008*; *Murphy et al., 2019*; *Murphy et al., 2018*; *Vatansever et al., 2017*). Extending these ideas, we recently showed that the DMN plays a role in motor adaptation, showing that connectivity between DMN regions and sensorimotor cortex is altered when normal visual-motor contingencies governing behavior are interrupted over the course of several trials (*Gale et al., 2022*). Given that the DMN is hypothesized to exert influence on functional brain activity via its topographic positioning on cortex (*Smallwood et al., 2021*), understanding how the DMN supports task

behavior likely requires analytical techniques that allow for a characterization of whole-brain changes in functional architecture.

In the current study, we explored changes in the landscape of cortical and striatal activity during a reward-based motor task in which human participants learned to produce, through purely reinforcement feedback, a specific movement trajectory that was initially unknown to them. To characterize learning-related changes at the neural level, we leveraged advanced manifold learning approaches that provide a low-dimensional description of cortical activity (*Huntenburg et al., 2018*; *Margulies et al., 2016*; *Vos de Wael et al., 2020*). This approach builds on recent electrophysiological studies in macaques, demonstrating that high-dimensional neural population activity can be described along a low-dimensional subspace or manifold (*Cunningham and Yu, 2014*; *Gallego et al., 2017*; *Shenoy et al., 2013*; *Vyas et al., 2020*), reflecting covariance patterns across the entire population. This same organizational structure also appears to govern the macroscale activity of cortex, with this manifold approach having recently provided key insights into the overarching structural and functional architecture of the human brain (*Huntenburg et al., 2018*; *Paquola et al., 2019*; *Shine et al., 2019a*; *Shine et al., 2019b*; *Vázquez-Rodríguez et al., 2019*). Here, we applied this manifold approach to explore how brain activity across widely distributed cortical and striatal systems is coordinated during reward-based motor learning. We were particularly interested in characterizing how connectivity between regions within the DMN and the rest of the brain changes as participants shift from learning the relationship between motor commands and reward feedback, during early learning, to subsequently using this information, during late learning. We were also interested in exploring whether learning-dependent changes in manifold structure relate to variation in subject motor performance.

## Results

Prior studies examining the neural processes underlying reward-based learning have typically used tasks requiring simple motor responses, such as button presses or lever movements (*Averbeck and O'Doherty, 2022*; *Daw et al., 2006*; *Klein-Flügge et al., 2022*; *Lee et al., 2012*; *O'Doherty et al., 2017*; *Rushworth et al., 2011*). The simplicity of these motor responses is intended to isolate participants' choice behavior by eliminating any variability related to movement execution (i.e., motor implementation of the choice) as a potential confounding factor in the learning process (*McDougle et al., 2019*; *McDougle et al., 2016*). However, recent theories on learning, supported by both human and animal studies (*Dhawale et al., 2019*; *Dhawale et al., 2017*; *Wu et al., 2014*), have highlighted the crucial role of movement variability—and motor exploration in particular—as a key ingredient for effective learning (*Dam et al., 2013*; *Wilson et al., 2021*; *Wu et al., 2014*). In order to incorporate this aspect to learning, and inspired by recent work in the field (*Dam et al., 2013*; *Wu et al., 2014*), we developed an MRI-compatible reward-based motor task in which human participants (N = 36) learned to shape their hand trajectories purely through reinforcement feedback.

In this task, subjects used their right finger on an MRI-compatible touchpad to trace, without visual feedback of their finger, a rightward-curved path displayed on a screen (*Figure 1A and B*). Participants began the MRI study by performing a *Baseline* block of 70 trials, wherein they did not receive any feedback about their performance. Following this, subjects began a separate *Learning* block of 200 trials in which they were told that they would now receive score feedback (from 0 to 100 points), presented at the end of each trial, based on how accurately they traced the visual path displayed on the screen. However, unbeknownst to subjects, the score they actually received was based on how well they traced a *hidden* mirror-image path (the 'reward' path, which was reflected across the vertical axis; *Figure 1C*). Importantly, because subjects received no visual feedback about their actual finger trajectory and could not see their own hand, they could only use the score feedback—and thus only reward-based learning mechanisms—to modify their movements from one trial to the next (*Dam et al., 2013*; *Wu et al., 2014*). That is, subjects could not use error-based learning mechanisms to achieve learning in our study as this form of learning requires sensory errors that convey both the change in direction and magnitude needed to correct the movement.

*Figure 1C* shows an example of a single subject's finger trajectories across trials. Initially, the subject begins by tracing the visual path displayed on the screen (as instructed), albeit with some expected motor noise due to the absence of any visual feedback about their finger paths (see cyan trajectories). However, over time, the subject learns to gradually trace a path more similar to the rewarded, mirror-image path (dark pink trajectories). As can be seen in *Figure 1D*, subjects on average were able to

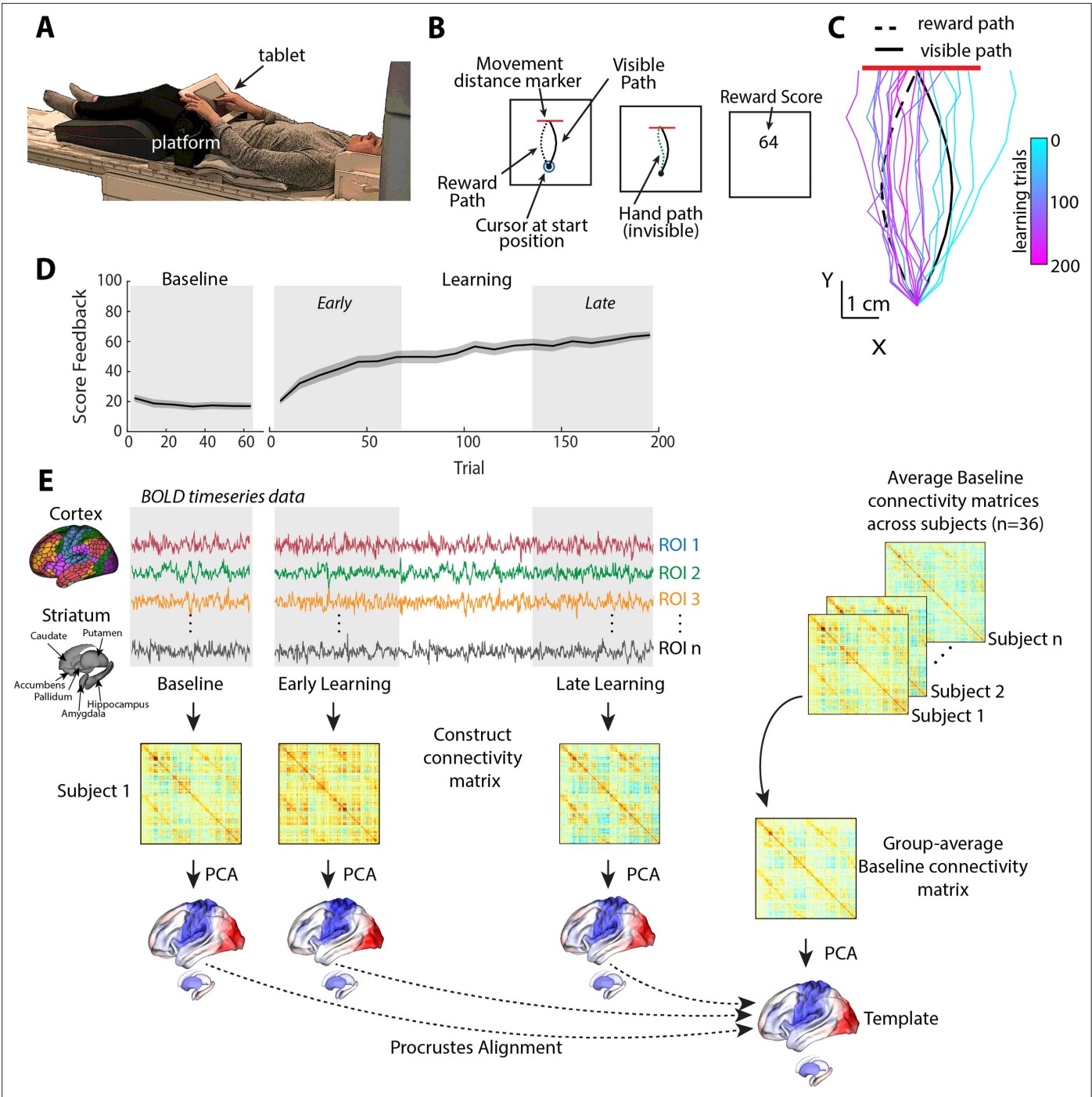

**Figure 1.** Task structure and overview of fMRI analysis. (**A**) Subject setup in the MRI scanner. (**B**) Trial structure of the reward-based motor learning task. On each trial, subjects were required to trace a curved (Visible) path from a start location to a target line (in red), without visual feedback of their finger location. Following a baseline block of trials, subjects were instructed that they would receive score feedback, presented at the end of the trial, based on their accuracy in tracing the visible path. However, unbeknownst to subjects, the score they received was actually based on how accurately they traced the mirror-image path (reward path), which was invisible to participants. (**C**) Example subject data from learning trials in the task. Colored traces show individual trials over time (each trace is separated by ten trials to give a sense of the trajectory changes over time; 20 trials shown in total). (**D**) Average participant performance throughout the learning task. Black line denotes the mean across participants whereas the gray banding denotes ±1 standard error of the mean (SEM). Three equal-length task epochs for subsequent neural analyses are indicated by the gray shaded boxes. (**E**) Neural analysis approach. For each participant and each task epoch (baseline, early, and late learning), we estimated functional connectivity matrices using region-wise time series extracted from the Schaefer 1000 cortical parcellation and the Harvard-Oxford striatal parcellation. We estimated functional connectivity manifolds for each task epoch using principal component analysis (PCA) with centered and thresholded connectivity matrices

*Figure 1 continued on next page*

*Figure 1 continued*

(see 'Materials and methods', as well as *Figure 2*). All manifolds were aligned to a common template manifold created from a group-average baseline connectivity matrix (right) using Proscrustes alignment. This allowed us to assess learning-related changes in manifold structure from this baseline architecture.

The online version of this article includes the following figure supplement(s) for figure 1:

**Figure supplement 1.** Behavioral measures of learning across the task.

**Figure supplement 2.** Variability in learning across subjects.

use the reward-based feedback to increase their score, and thus produce a trajectory more similar to the hidden path, over the 200 learning trials (see *Figure 1—figure supplement 1* for other measures of changes in subject motor behavior throughout learning, and *Figure 1—figure supplement 2* for examples of all subjects' finger trajectories across trials).

In order to study the changes in functional cortical and striatal organization during the learning task, we used three distinct, equal-length epochs over the time course of the study. Specifically, in addition to the task baseline epoch (70 trials), we defined early and late learning epochs as the subsequent initial and final 70 trials, respectively, following the presentation onset of reward feedback. For each participant, we extracted mean blood oxygenation level-dependent (BOLD) time-series data for each cortical region defined by the Shaefer 1000 cortical parcellation (*Schaefer et al., 2018*) and for striatal regions defined by the Harvard-Oxford parcellation (*Avants et al., 2008*; *Frazier et al., 2005*), and then estimated covariance (functional connectivity) matrices for each epoch (baseline, early, and late; *Figure 1E*; for a similar approach, see *Gale et al., 2022*).

Because prior work (*Gordon et al., 2017*; *Gratton et al., 2018*), including our own (*Areshenkoff et al., 2022*; *Areshenkoff et al., 2021*; *Gale et al., 2022*), suggests that individual differences in functional connectivity can obscure any task-related effects, we centered the connectivity matrices using the Riemannian manifold approach (*Areshenkoff et al., 2022*; *Areshenkoff et al., 2021*; *Gale et al., 2022*; *Zhao et al., 2018*; see *Figure 2—figure supplement 1* for an overview of the approach). To illustrate the effects of this centering procedure, and why it is important for elucidating task-related effects in the data, we projected participants' covariance matrices both prior to, and after the centering procedure, using uniform manifold approximation (UMAP; *McInnes et al., 2018*). As shown in *Figure 2A*, prior to the centering procedure the covariance matrices mainly cluster according to subject identity, consistent with prior findings showing that this subject-level structure explains the majority of the variance in functional connectivity data (*Gratton et al., 2018*). Clearly, this subject-level clustering could impact the ability to detect task-related effects in the data. However, after applying the centering procedure (*Figure 2B*), this subject-level clustering is abolished, potentially allowing for the differentiation of the three task-related epochs.

To examine the changes in cortical and striatal connectivity during the reward-based motor learning task, we used the centered matrices from *Figure 2B* to estimate separate cortical-striatal connectivity manifolds for each participant's baseline, early, and late covariance matrices (see also *Gale et al., 2022*). Following from prior work (*Hong et al., 2020*; *Paquola et al., 2019*; *Vos de Wael et al., 2020*), we transformed each matrix into an affinity matrix and then applied principal components analysis (PCA) to obtain a set of principal components (PCs) that provides a low-dimensional representation of cortical-striatal functional organization (i.e., a cortical-striatal manifold). Next, we aligned the manifolds from each participant to a template baseline manifold, which we constructed using the mean of all baseline connectivity matrices across participants (*Figure 1E*). We did this for two reasons: (1) the baseline manifold provided a common target for manifold alignment (*Vos de Wael et al., 2020*) so that all subjects could be directly compared in a common task-based neural space, and (2) it allowed us to selectively detect deviations *from this* baseline manifold architecture; that is, observe the changes to this manifold structure that occur as a function of learning during the task (when subjects begin receiving reward feedback about their performance).

## Cortical-striatal manifold structure during baseline trials

The top three PCs of the template baseline manifold (*Figure 3A*) describe the cortical-striatal functional organization during baseline trials. As can be seen in *Figure 3A*, PC1 distinguishes visual regions (positive loadings in red) from somatomotor regions (negative loadings in blue). Meanwhile, PC2

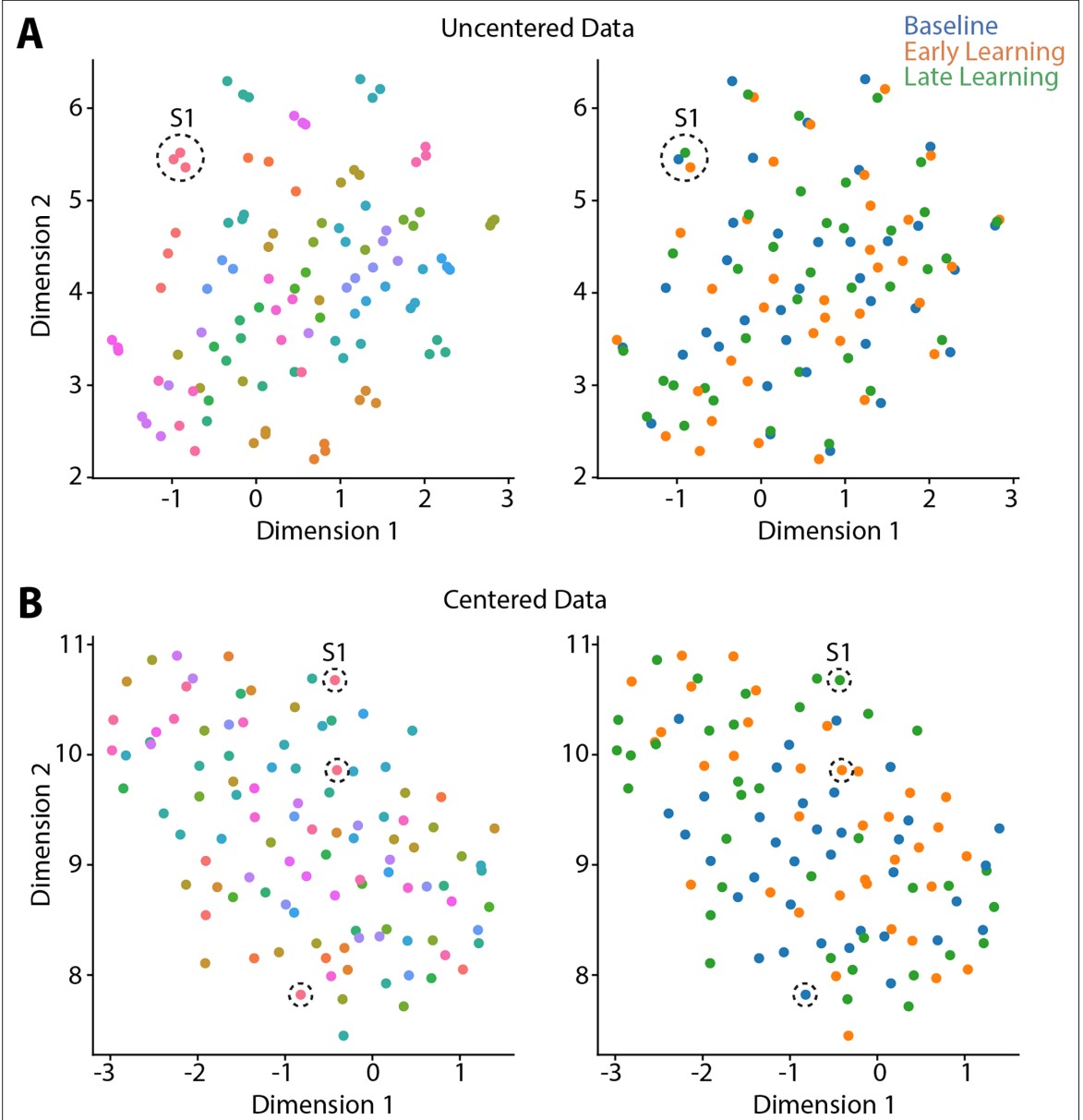

**Figure 2.** Riemmanian centering removes subject-level clustering. Uniform Manifold Approximation (UMAP) visualization of the similarity of connectivity matrices, both before centering (**A**) and after (**B**) centering. In these plots, each point represents a single functional connectivity matrix, color-coded either to subject identity (left panels) or task epoch (right panels), with its location in the multidimensional space based on the similarity between matrices. Note that the uncentered connectivity matrices in (**A**) show a high degree of subject-level clustering, thus obscuring any differences in task structure. By contrast, the Riemmanian manifold centering approach (in **B**) abolishes this subject-level clustering. To help illustrate this point, the dashed circles in both (**A**) and (**B**) indicate the functional connectivity matrices belonging to the same single subject (subject 1; S1).

The online version of this article includes the following figure supplement(s) for figure 2:

**Figure supplement 1.** Overview of the Riemannian manifold centering approach.

distinguishes visual and somatomotor regions (in red) from the remaining cortical areas (in blue), most prominently high-order association regions within the DMN. Finally, PC3 mainly constitutes a gradient of frontoparietal areas of the dorsal attention network (DAN) and frontoparietal control network (FCN) versus DMN regions. Collectively, these top three PCs explain ~70% of the total variance (*Figure 3B*).

When we mapped the brain regions onto their assigned intrinsic functional network architecture (*Thomas Yeo et al., 2011*), we confirmed that PCs 1 and 2 jointly differentiate visual, DMN, and somatomotor regions, replicating the tripartite structure of the brain's intrinsic functional architecture

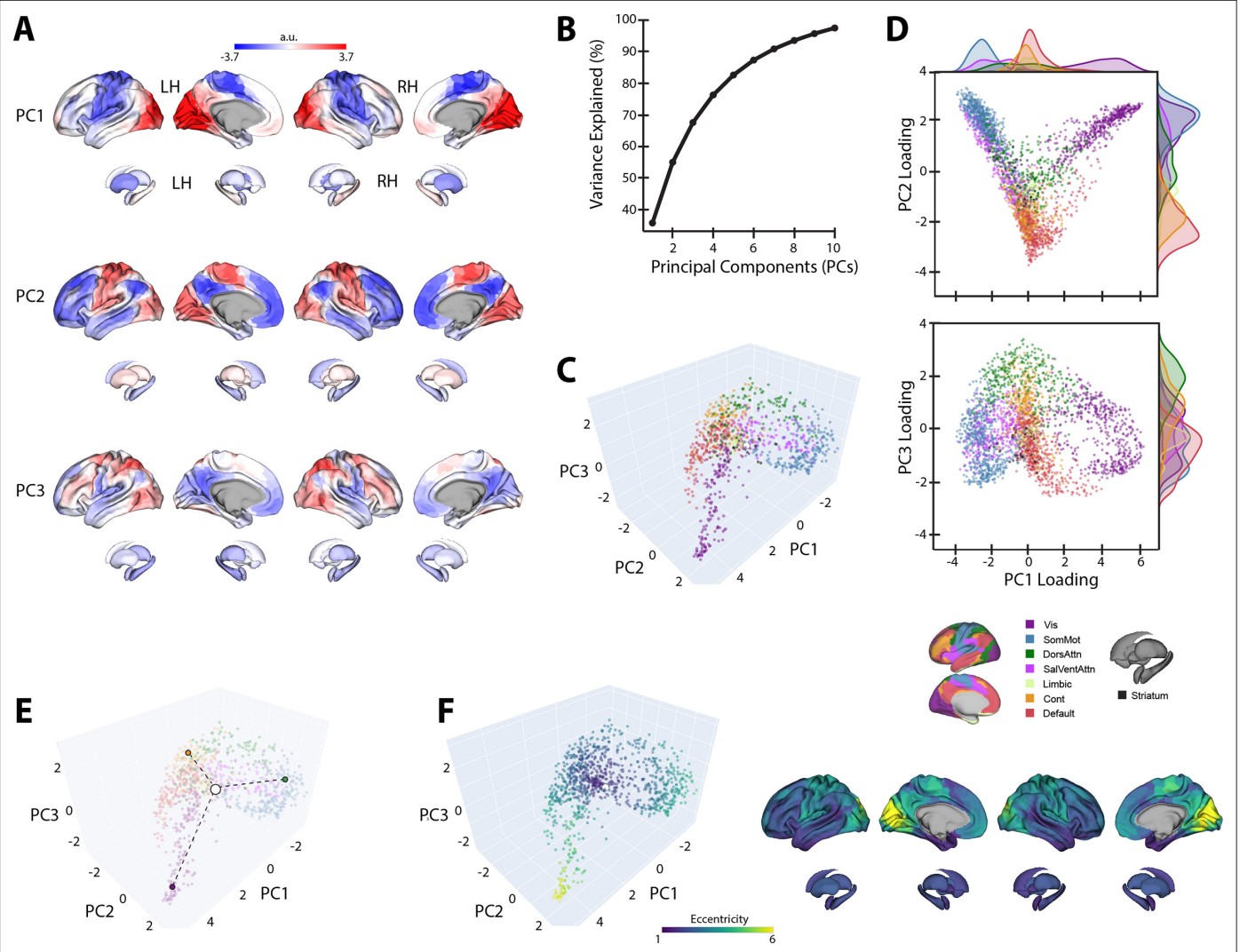

**Figure 3.** Baseline manifold structure and eccentricity. (**A**) Region loadings for the top three principal components (PCs). (**B**) Percent variance explained for the first 10 PCs. (**C**) The baseline (template) manifold in low-dimensional space, with regions colored according to functional network assignment (*Schaefer et al., 2018*; *Thomas Yeo et al., 2011*). (**D**) Scatter plots showing the embedding of each region along the top three PCs. Probability density histograms at the top and right show the distribution of each functional network along each PC. Vis: visual; SomMot: somatomotor; DorsAttn: dorsal attention; SalVentAttn: salience/ventral attention; Cont: control. (**E**) Illustration of how eccentricity is calculated. Region eccentricity along the manifold is computed as the Euclidean distance (dashed line) from manifold centroid (white circle). The eccentricity of three example brain regions is highlighted (bordered colored circles). (**F**) Regional eccentricity during baseline. Each brain region's eccentricity is color-coded in the low-dimensional manifold space (left) and on the cortical and striatal surfaces (right). White circle with black bordering denotes the center of the manifold (manifold centroid).

The online version of this article includes the following figure supplement(s) for figure 3:

**Figure supplement 1.** Functional connectivity properties that underlie manifold eccentricity.

**Figure supplement 2.** Derivation of cortical gradients did not depend on the inclusion of the striatum in the analysis.

---

(*Huntenburg et al., 2018*; *Margulies et al., 2016*; *Figure 3D*). Others have argued that this tripartite structure is a fundamental feature of functional brain organization, whereby the transition from unimodal cortex (visual and somatomotor networks) to transmodal cortex (the DMN) reflects a global processing hierarchy from lower- to higher-order brain systems (*Huntenburg et al., 2018*; *Margulies et al., 2016*; *Smallwood et al., 2021*).

We next sought to characterize the relative positions of cortical and striatal brain regions along the baseline connectivity-derived manifold space, thus providing a basis to examine future changes in the positioning of these regions during early and late learning. To this aim, and following from previous methods (*Gale et al., 2022*; *Park et al., 2021a*), we computed the manifold eccentricity of each

region by taking its Euclidean distance from manifold centroid (coordinates (0,0,0); see *Figure 3E*). This eccentricity measure provides a multivariate index of each brain region's embedding in the three-dimensional manifold space, whereby distal regions located at the extremes of the manifold have greater eccentricity than proximal regions located near the manifold center (*Figure 3E*). Under this framework, regions with higher eccentricity are interpreted as having higher functional segregation from other networks in the rest of the brain, whereas regions with lower eccentricity are interpreted as having higher integration (lower segregation) with other networks in the rest of the brain (*Park et al., 2021a*; *Park et al., 2021b*; *Valk et al., 2023*). Consistent with this interpretation, we find that our eccentricity measure strongly correlates with various graph theoretical measures of integration and segregation. For instance, we find that baseline eccentricity is positively related to cortical node strength ($r = 0.88$, two-tailed $p<0.001$) and within-manifold degree z-score ($r = 0.45$, two-tailed $p<0.001$), consistent with the notion that more eccentric regions are more strongly functionally coupled with other members of the same functional network (i.e., higher segregation; see *Figure 3—figure supplement 1*). Likewise, we find that baseline eccentricity is negatively related to a region's participation coefficient ($r = -0.74$, two-tailed $p<0.001$), which is a measure of a region's degree of cross-network integration. Thus, taken together, changes in a brain region's eccentricity can provide us with a multivariate measure of changes in that region's functional segregation versus integration during early and late learning.

## Changes in cortical-striatal manifold structure during learning

To examine which regions exhibited significant changes in manifold eccentricity from (1) baseline to early learning and then from (2) early to late learning, we performed two sets of paired *t*-tests and corrected for multiple comparisons using a false discovery rate correction (FDR; q < 0.05). To directly test how regional eccentricity changes at the onset of learning (when subjects begin receiving reward feedback), we performed a contrast of early > baseline (*Figure 4A*). This contrast primarily revealed a pattern of increasing eccentricity, that is, manifold expansion, across several brain regions, indicating that these regions became segregated from the rest of the brain (red areas in *Figure 4A*). This included areas located throughout the cortical sensorimotor system and DAN, including bilateral superior-parietal, somatomotor, supplementary motor, and premotor cortex, as well as regions in lateral visual cortex. In addition, this contrast identified many key areas of the DMN, including bilateral medial frontal gyrus (MFG), MPFC, inferior frontal gyrus (IFG), and middle temporal cortex (MTC; for a network-level summary of these general effects, see the spider plot in *Figure 4A*). In contrast to this general pattern of expansion-related effects, we also found that a small subset of areas in the posterior medial cortex (PMC) and posterior angular gyrus (AG) instead exhibited a decrease in eccentricity, that is, manifold contraction (regions in blue in *Figure 4A*), indicating that these areas increased their integration with other areas of the brain. Notably, we did not observe any significant changes in striatal regions from baseline to early learning (however, for interested readers, *Figure 4—figure supplement 1* shows the unthresholded data from both cortex and striatum to indicate any trends).

Next, to examine how regional eccentricity changes over the course of learning, we performed a direct contrast of late > early learning (*Figure 4B*). This contrast mainly revealed a reversal in the general pattern of the effects observed in the DMN during early learning. Specifically, during late learning, several regions in bilateral MFG, MPFC, IFG, and MTC now exhibited contraction along the manifold, indicating an increased integration of these areas with other regions of the brain (this reversal can be easily observed by comparing the red areas in *Figure 4A* to the blue areas in *Figure 4B*). By contrast, areas in PMC and posterior AG now exhibited expansion, indicating an increased segregation of these areas from the brain. In addition, during late learning, we observed manifold expansion in several areas of the salience/ventral attention network (SalVentAttn), including the dorsal anterior cingulate cortex (dACC) and the anterior insula (AI), as well as higher-order lateral occipital cortical areas, and areas in retrosplenial cortex and medial ventro-temporal cortex (a network-level summary of these effects can be found in the *Figure 4A/B* spider plot). Again, as in the early > baseline contrast, we did not observe any significant changes in striatal regions from early to late learning. The only region that came close to reaching statistical significance in the striatum was the right pallidum ($p=0.01$), but this region did not pass whole-brain FDR correction (corrected alpha = 0.086; note that *Figure 4—figure supplement 1* shows the unthresholded maps for this contrast to demonstrate the strong reversal in the pattern of effects during late learning, as well as indicate any trends).

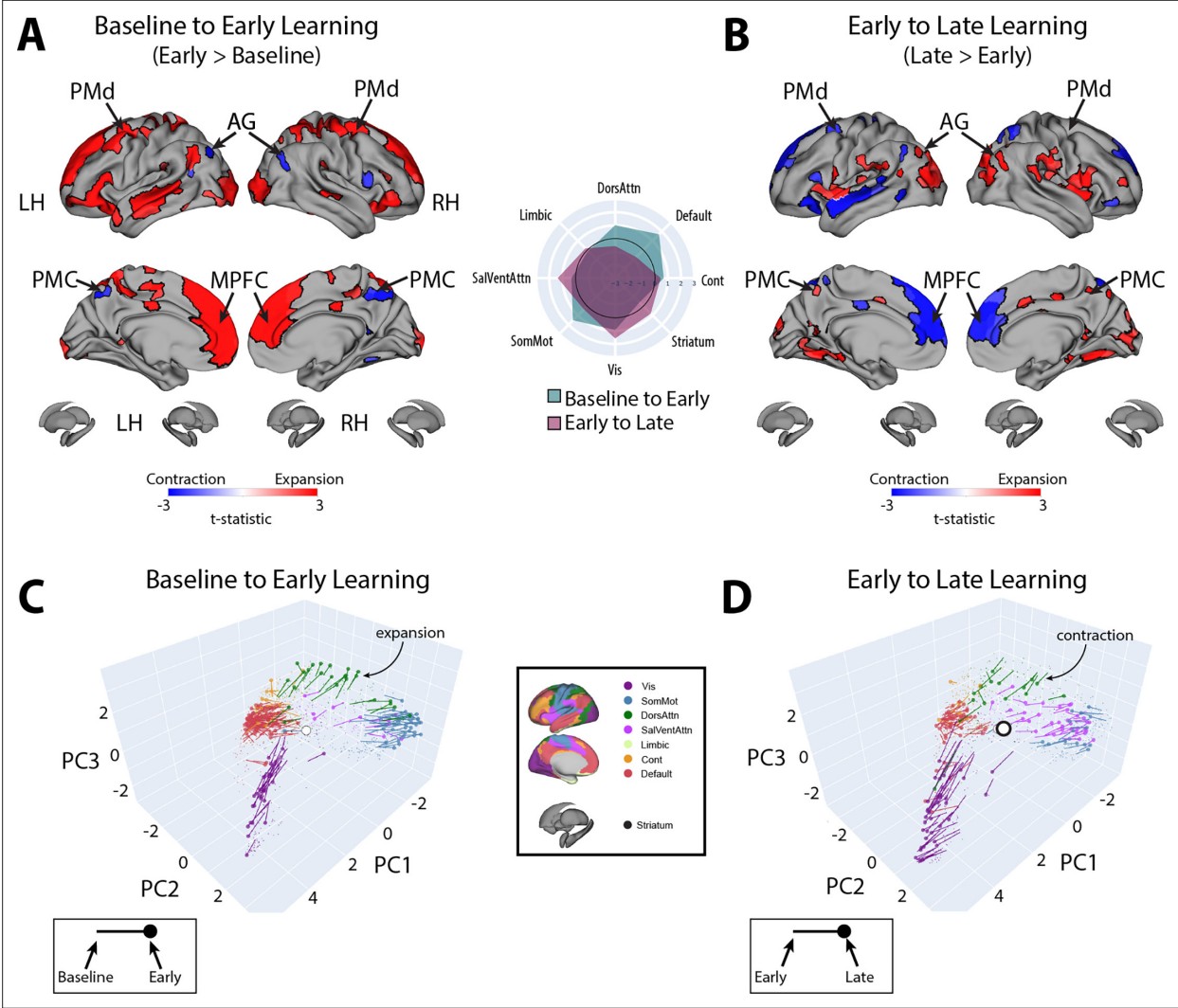

**Figure 4.** Changes in manifold structure during reward-based motor learning. (**A, B**) Pairwise contrasts of eccentricity between task epochs (N=36). Positive (red) and negative (blue) values show significant increases and decreases in eccentricity (i.e., expansion and contraction along the manifold), respectively, following false discovery rate (FDR) correction for region-wise paired *t*-tests (at q < 0.05). The spider plot, at center, summarizes these patterns of changes in connectivity at the network-level (according to the Yeo networks, ***Thomas Yeo et al., 2011***). Note that the black circle in the spider plot denotes *t* = 0 (i.e., no change in eccentricity between the epochs being compared). Radial axis values indicate *t*-values for the associated contrast (see color legend). (**C, D**) Temporal trajectories of statistically significant regions from (**A**) and (**B**), shown in the low-dimensional manifold space. Traces show the displacement of each region for the relevant contrast and filled colored circles indicate each region's final position along the manifold for a given contrast (see insets for legends). Each region is colored according to its functional network assignment (middle). Nonsignificant regions are denoted by the gray point cloud. White circle with black bordering denotes the center of the manifold (manifold centroid).

The online version of this article includes the following figure supplement(s) for figure 4:

**Figure supplement 1.** Unthresholded maps of changes in manifold structure during early and late learning.

**Figure supplement 2.** Changes in manifold structure from baseline to late learning.

**Figure supplement 3.** Changes in manifold structure during reward-based motor learning for the Yeo 17 network parcellation.

Finally, for completeness, we also examined the contrast of late > baseline, which solely revealed a pattern of cortical expansion across several regions—in particular in areas of the SalVentAttn and visual network (see ***Figure 4—figure supplement 2***). This indicates a continuing expansion (and segregation) of these regions as learning progresses.

Taken together, the above pattern of results suggests that, during early learning, transmodal areas of the DMN, as well as several areas of the sensorimotor system (including areas of the DAN), begin to segregate from other brain networks, whereas a subset of areas—the PMC and posterior

AG in particular—begin to integrate with regions outside of their respective networks. By contrast, during late learning, there is a clear reversal in these patterns, with regions within the DMN and DAN beginning to integrate with areas belonging to other brain networks. In the next section, we directly examine these interpretations of manifold expansion and contraction during early and late learning.

## Changes in connectivity that underlie patterns of manifold reconfiguration

Given that eccentricity provides a multivariate index of a region's overall profile of connectivity (i.e., its relative positioning on the manifold), we next performed seed connectivity analyses to further characterize the patterns of effects that underlie the expansions and contractions of manifold structure during learning. To this aim, we selected several representative regions, distributed throughout the cortex, that epitomize the main changes in eccentricity that we observed during early learning (shown in *Figure 4A*). These regions included the left (contralateral) MPFC, PMd, and PMC, allowing us to characterize the patterns of connectivity changes across prefrontal, premotor, and parietal cortex, respectively (for seed-connectivity analyses of their right hemisphere homologs, see *Figure 5—figure supplement 1*). For each region, we contrasted seed connectivity maps between both the early learning vs. baseline epochs (early > baseline) and the late vs. early learning epochs (late > early) by computing region-wise paired *t*-tests, thus producing contrast maps associated with the connectivity change of each representative seed region (*Figure 5*). Note that in *Figure 5* we display the unthresholded voxel-wise contrast maps (two leftmost panels), the region's corresponding change in eccentricity across epochs (second from rightmost panel), and the corresponding spider plots depicting network-level changes (rightmost panel), thus allowing for a complete visualization of the collective changes in connectivity that contribute to the changes in regional eccentricity. (Note that, for the spider plots, we used the 17-network mapping in order to capitalize on the improved spatial precision compared to the 7-network mapping; *Schaefer et al., 2018*; *Thomas Yeo et al., 2011*.)

During early learning, we found that the left MPFC seed region, associated with the DMN network, exhibited increased connectivity with other DMN subregions and reduced connectivity with superior parietal and premotor areas in the DAN (*Figure 5A*). By contrast, during late learning, we observed a reversal in this pattern of connectivity changes, with the MPFC now exhibiting increased connectivity with the same regions of the DAN but reduced connectivity with other DMN areas. Together, these results suggest that the manifold expansion of the MPFC during early learning arises from its increased connectivity with other DMN areas (i.e., segregation of the DMN), whereas the manifold contraction of this region during late learning arises from its increased connectivity with areas *outside* of the DMN, such as sensorimotor areas of the DAN.

Notably, for the left PMd seed region (*Figure 5B*), associated with the DAN, we observed an inverse pattern of results from that observed for the MPFC region above. Specifically, during early learning, we observed increased connectivity of the left PMd with other areas of the DAN, as well as areas belonging to the SalVentAttn network, such as the anterior insula/IFG, dACC, and inferior parietal cortex. Notably, this was coupled with its decreased connectivity to DMN areas and the hippocampus (*Figure 5B*). By contrast, during late learning, we observed a reversal in this pattern of effects, whereby connectivity with DMN areas and the hippocampus now increased, whereas connectivity with the DAN and SalVentAttn areas decreased. In this case, the pattern of manifold expansion and contraction of PMd during early and late learning, respectively, likely arises from its increased connectivity with attention networks in brain (the DAN and SalVentAttn) during early learning and an increase in between-network connectivity (i.e., integration) with DMN areas during late learning.

Finally, for the left PMC seed region, located at the border of the DMN-A and Control-C networks—and that was one of the few regions that exhibited *contraction* during early learning—we found that this region exhibited decreased connectivity during early learning with other DMN-A and Control-C subregions in bilateral PMC, as well as decreased connectivity with bilateral hippocampus and MPFC (*Figure 5C*). Instead, this PMC region exhibited increased connectivity with DAN areas in superior parietal cortex and premotor cortex, and most prominently, with areas in the anterior insula/IFG, dACC, and inferior parietal cortex (belonging to the SalVentAttn networks). By contrast, during late learning, we again observed a reversal in this pattern of connectivity changes, with the PMC seed region now exhibiting increased connectivity with other bilateral PMC areas, MPFC and the hippocampus, as well as reduced connectivity with the same DAN and SalVentAttn areas. Together, these

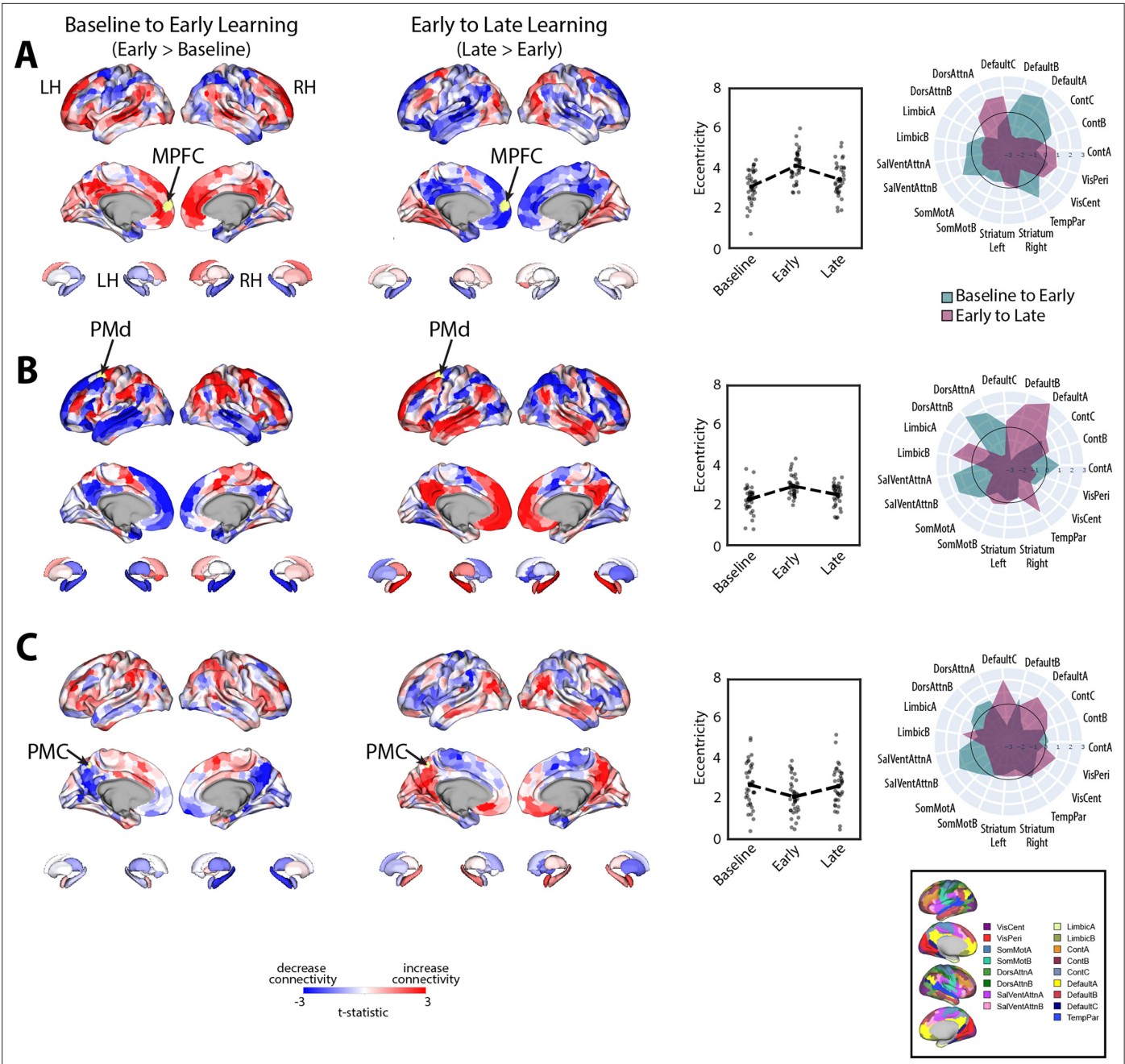

**Figure 5.** Main patterns of connectivity changes that underlie manifold expansions and contractions. (**A–C**) Connectivity changes for each seed region. Selected seed regions are shown in yellow and are also indicated by arrows. Positive (red) and negative (blue) values show increases and decreases in connectivity, respectively, from baseline to early learning (leftmost panel) and early to late learning (second from leftmost panel). Second from the rightmost panel shows the eccentricity of each region for each participant (N=36), with the black dashed line showing the group mean for each task epoch. Rightmost panel contains spider plots, which summarize these patterns of changes in connectivity at the network level (according to the Yeo 17-networks parcellation; *Thomas Yeo et al., 2011*). Note that the black circle in the spider plot denotes *t* = 0 (i.e., zero change in eccentricity between the epochs being compared). Radial axis values indicate *t*-values for associated contrast (see color legend).

The online version of this article includes the following figure supplement(s) for figure 5:

**Figure supplement 1.** Patterns of connectivity changes that underlie manifold expansions and contractions for right-hemisphere regions.

results suggest that manifold contractions of the PMC during early learning arise from its increased integration with regions outside of the DMN-A and Control-C networks, such as the DAN and SalVentAttn, whereas the manifold expansions of this region during late learning arise from its increased within-network connectivity with other DMN and Control areas (i.e., segregation).

Taken together, the results of our seed connectivity analyses above are broadly consistent with our interpretation of the patterns of manifold expansion as reflecting increases in *within*-network connectivity (segregation) and the patterns of manifold contraction as reflecting increases in *between*-network connectivity (integration). More generally, however, these findings point to changes in the landscape of communication between regions of the DAN, SalVentAttn, and DMN in particular, as being associated with reward-based motor learning. Specifically, we find that, during early learning, there is increased functional coupling between several sensorimotor areas of the DAN with areas of the SalVentAttn network, whereas during late learning, these DAN sensorimotor areas switch their connectivity to DMN areas.

## Changes in eccentricity relate to learning performance

In the previous sections, we characterized the patterns of mean changes in manifold structure during learning across all participants. However, it is well established that subjects exhibit significant variation in motor learning ability (*de Brouwer et al., 2018*; *de Brouwer et al., 2022*; *Standage et al., 2023*;

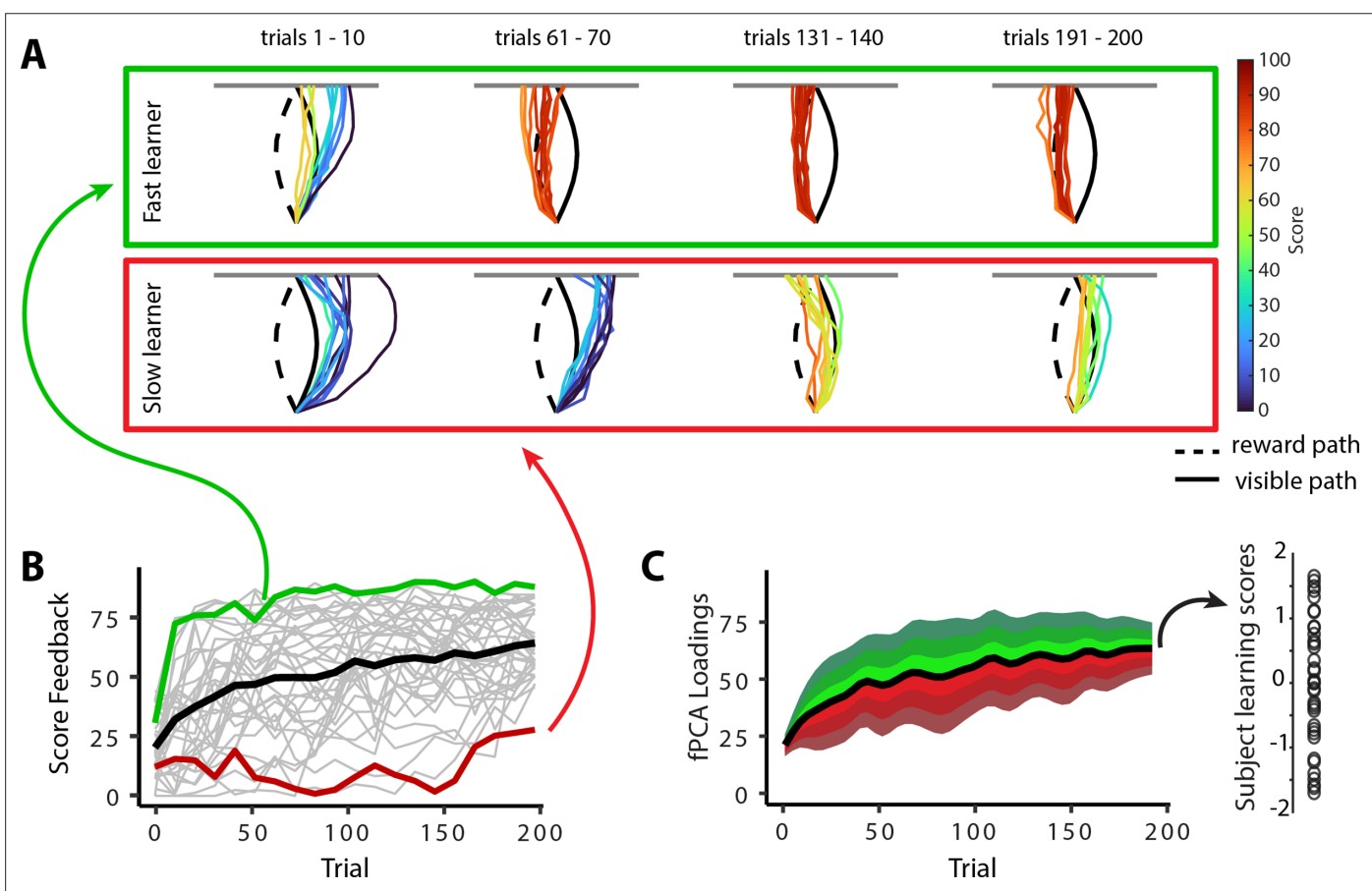

**Figure 6.** Individual differences in subject learning performance. (**A**) Examples of a good learner (bordered in green) and poor learner (bordered in red). (**B**) Individual subject learning curves for the task. Solid black line denotes the mean across all subjects (N=36), whereas light gray lines denote individual participants. The green and red traces denote the learning curves for the example good and poor learners denoted in (**A**). (**C**) Derivation of subject learning scores. We performed functional principal component analysis (fPCA) on subjects' learning curves in order to identify the dominant patterns of variability during learning. The top component, which encodes overall learning, explained the majority of the observed variance (~75%). The green and red bands denote the effect of positive and negative component scores, respectively, relative to mean performance. Thus, subjects who learned more quickly than average have a higher loading (in green) on this 'learning score' component than subjects who learned more slowly (in red) than average. The plot at the right denotes the loading for each participant (open circles) onto this learning score component.

*Wu et al., 2014*). Indeed, while the learning curve in *Figure 1D* shows that subjects, on average, improved their scores during the task, this group-level result obscures the fact that individuals differed greatly in their rates of learning (see individual subject learning curves in *Figure 1—figure supplement 2*). To emphasize this fact, *Figure 6A* highlights the learning curves for two example subjects: an individual who learned the hidden shape quite rapidly (a 'fast learner' in green) and a second individual who only gradually learned to trace the hidden shape (a 'slow learner' in red). To quantify this variation in subject performance in a manner that accounted the auto-correlation in learning performance over time (i.e., subjects who learned more quickly tend to exhibit better performance by the end of learning), we opted for a pure data-driven approach and performed functional principal component analysis (fPCA; *Shang, 2014*) on subjects' learning curves. This approach allowed us to isolate the dominant patterns of variability in subject's learning curves over time (see 'Materials and methods' for further details; see also *Areshenkoff et al., 2022*). Using this fPCA approach, we found that a single component—encoding overall learning—captured the majority (~75%) of the variability in subjects' learning curves (*Figure 6C*). We thus used each subjects' loading on this dominant component as a single scalar measure of subjects' overall learning performance: Individuals who tended to learn the task more quickly had higher values on this 'learning score' component, whereas individuals who tended to learn the task more slowly had lower values on this component (see single data points in *Figure 6C* at the right).

Next, to examine the neural correlates of intersubject differences in learning scores, we calculated, for each region, the association between participants' scores and the change in eccentricity between the baseline and early learning epochs (early > baseline; *Figure 7A*). This analysis did not reveal any brain regions that survived FDR corrections for multiple comparisons (q < 0.05). However, the FDR approach is completely agnostic to any topographical patterns of effects across brain areas, which may correspond with known functional networks. Indeed, the full cortex and striatal correlation map in *Figure 7A* indicates that many region-level correlations exhibit a high degree of spatial contiguity, with many statistically significant regions (at p<0.05, bordered in black) being situated within much larger clusters of regions that exhibit a similar pattern of effects (i.e., areas in blue, denoting a negative correlation between learning score versus the change in eccentricity from baseline to early learning, tend to lie adjacent to other regions exhibiting a similar negative correlation). This is because topographically adjacent regions are likely to have similar connectivity profiles, and thus project onto similar locations along the manifold, resulting in similar brain–behavior relationships. This spatial topography suggests that the association between the eccentricity of certain brain regions and learning performance are likely to be better characterized at the level of distributed functional networks.

To examine this, we mapped each region onto its respective functional network using the Yeo et al. 17-network parcellation (*Thomas Yeo et al., 2011*) and, for each participant, computed the mean manifold eccentricity for each network (i.e., network eccentricity). We then correlated the change in each brain network's eccentricity across epochs with subject learning scores. We tested the statistical significance of these network-level correlations by building null models that account for the spatial autocorrelation in the brain maps (*Markello et al., 2022*; *Váša et al., 2018*; see 'Materials and methods') and corrected for multiple comparisons (across all networks) using an FDR correction (q < 0.05). Using this permutation testing approach, we found that it was only the change in eccentricity of the DAN-A network that correlated with learning score (see *Figure 7C*), such that the more the DAN-A network *decreased* in eccentricity from baseline to early learning (i.e., contracted along the manifold), the better subjects performed at the task (see *Figure 7C*, scatterplot at the right). Consistent with the notion that changes in the eccentricity of the DAN-A network are linked to learning performance, we also found the inverse pattern of effects during late learning, whereby the more that this same network *increased* in eccentricity from early to late learning (i.e., expanded along the manifold), the better subjects performed at the task (*Figure 7D*). We should note that this pattern of performance effects for the DAN-A—that is, greater contraction during early learning and greater expansion during late learning being associated with better learning—appears at odds with the group-level effects described in *Figure 4A and B*, where we generally find the opposite pattern for the entire DAN network (composed of the DAN-A and DAN-B subnetworks). However, this potential discrepancy can be explained when examining the changes in eccentricity using the 17-network parcellation (see *Figure 4—figure supplement 3*). At this higher resolution level, we find that these

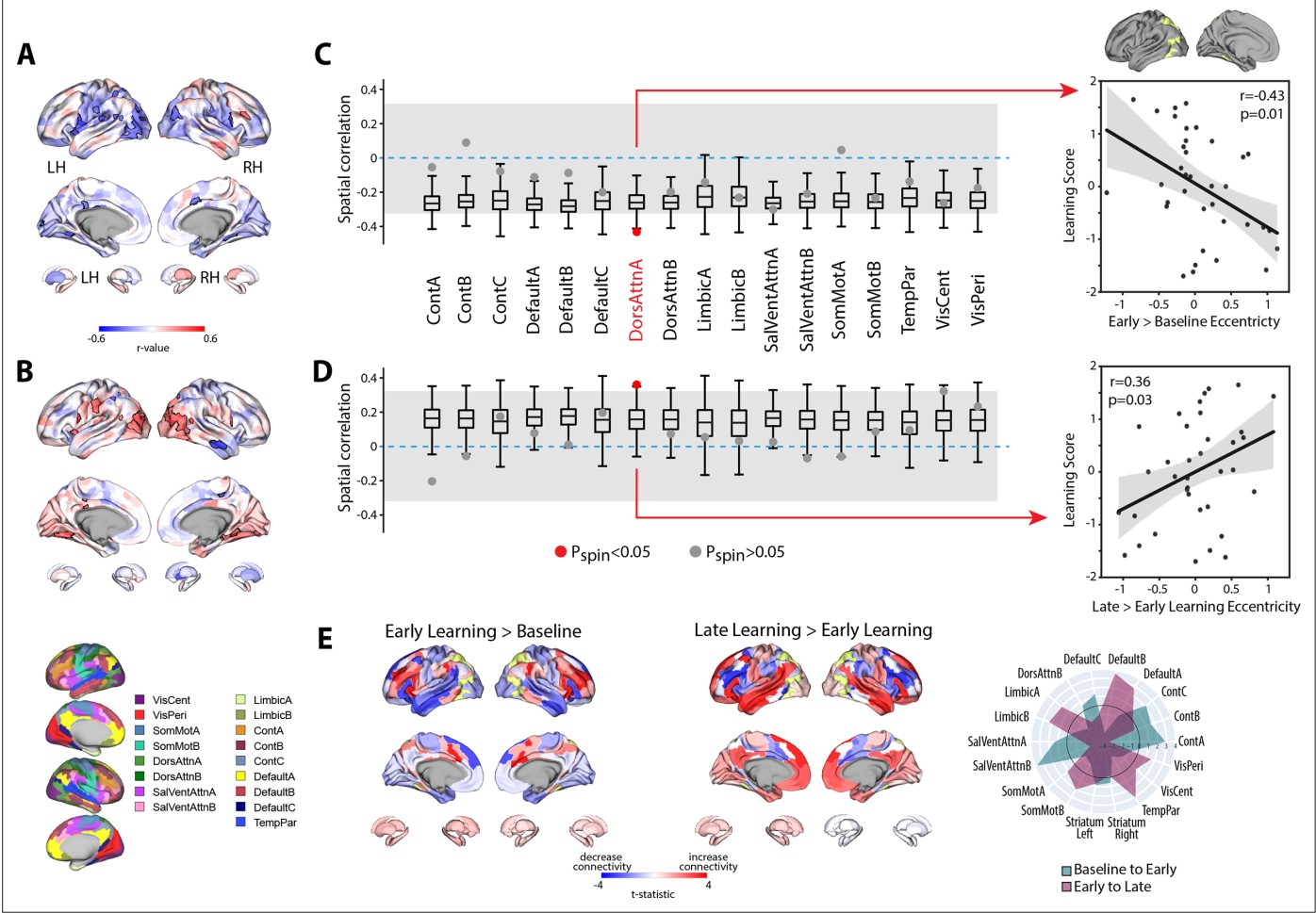

**Figure 7.** Relationship between learning performance and regional changes in eccentricity. (**A, B**) Whole-brain correlation map between subject learning score and the change in regional eccentricity from baseline to early learning (**A**) and early to late learning (**B**). Black bordering denotes regions that are significant at p<0.05. (**C, D**) Results of the spin-test permutation procedure, assessing whether the topography of correlations in (**A**) and (**B**) are specific to individual functional brain networks. Single points indicate the real correlation value for each of the 17 Yeo et al. networks (**Thomas Yeo et al., 2011**), whereas the boxplots represent the parameters of a null distribution of correlations derived from 1000 iterations of a spatial autocorrelation-preserving null model (**Markello et al., 2022**; **Váša et al., 2018**). In the boxplots, the ends of the boxes represent the first (25%) and third (75%) quartiles, the center line represents the median, and the whiskers represent the min-max range of the null distribution. All correlations were corrected for multiple comparisons (q < 0.05). The dashed horizontal blue line indicates a correlation value of zero and the gray shading encompasses correlation values that do not significantly differ from zero (p>0.05). (Note that in the spin-test procedure, due to the sign of the correlations, it is possible for some networks to be significantly different from the null distribution, and yet not significantly different from zero. Thus, to be considered significant in our analyses, a brain network was required to satisfy both constraints; i.e., show a correlation that is significantly different from zero *and* from the spatial null distribution.) Right, scatterplots show the relationships between subject learning score and the change in eccentricity from baseline to early learning (top) and early to late learning (bottom) for the DAN-A network (depicted in yellow on the cortical surface at top), the only brain network to satisfy the two constraints of our statistical testing procedure. Black line denotes the best-fit regression line, with shading indicating ±1 SEM. Dots indicate single participants (N=36). (**E**) Connectivity changes for the DAN-A network (highlighted in yellow) across epochs. Positive (red) and negative (blue) values show increases and decreases in connectivity, respectively, from baseline to early learning (left panel) and early to late learning (right panel). Spider plot, at the right, summarizes the patterns of changes in connectivity at the network-level. Note that the black circle in the spider plot denotes *t* = 0 (i.e., no change in eccentricity between the epochs being compared). Radial axis values indicate *t*-values for associated contrast (see color legend). VisCent: visual central; VisPer: visual peripheral; SomMotA: somatomotor A; SomMotB: somatomotor B; TempPar: temporal parietal; DorsAttnA: dorsal attention A; DorsAttnB: dorsal attention B; SalVentAttnA: salience/ventral attention A; SalVentAttnB: salience/ventral attention B; ContA: control A; ContB: control B; ContC: control C.

group-level effects for the entire DAN network are being largely driven by eccentricity changes in the DAN-B network (areas in anterior superior parietal cortex and premotor cortex), and not by mean changes in the DAN-A network. By contrast, our present results suggest that it is the contraction and expansion of areas of the DAN-A network (and not DAN-B network) that are selectively associated with subject learning performance.

To understand the global changes in connectivity that underlie these network eccentricity effects, we performed a network-level seed connectivity analyses (analogous to our single ROI seed connectivity analyses in the previous section) wherein constructed contrast maps, using the DAN-A as the seed network, for both the early vs. baseline epochs (early > baseline) and the late vs. early epochs (late > early). As before, we display the unthresholded voxel-wise contrast maps, along with corresponding spider plots depicting the network-level changes (*Figure 7E*), to allow for a complete visualization of the collective changes in network-level connectivity that underlie the changes in eccentricity of the DAN-A. As can be observed in *Figure 7E*, we find that, during early learning, DAN-A regions exhibited the largest increases in connectivity with one of the SalVentAttn subnetworks (SalVentAttn-B), whereas, during late learning, the DAN-A regions exhibited the largest increases in connectivity with one of the DMN subnetworks (DMN-B). These findings not only re-constitute the group-level effects reported above at the ROI level (in *Figure 5*), but they also suggest that this general transition in functional coupling—between the DAN and SalVentAttn areas during early learning, to DAN and DMN areas during late learning—is associated with differences in subject performance.

## Discussion

Complex behavior necessitates the coordinated activity of multiple specialized neural systems distributed across cortex and striatum. During motor learning, these systems must adapt their functional interactions to ensure appropriate behavior in response to changes in sensory feedback. While much research in motor learning has focused on understanding the role of sensorimotor cortex in isolation, our understanding of the contribution of higher-order brain systems, which play a role in the organization of behavior over time, remains incomplete. In the current study, we utilized state-of-the-art analytical methods that reconcile topographic and functional brain organization, enabling us to describe the changes in the landscape of cortical and striatal activity during learning.

During early learning, when subjects were establishing the relationship between motor commands and reward feedback, we found that regions within both the DAN (e.g., premotor cortex) and DMN (e.g., MPFC) exhibited expansion along the manifold. Our connectivity analyses revealed that this expansion was largely driven by an increase in within-network communication in both the DAN and DMN networks. There were, however, two notable exceptions to this general pattern. First, we found that connectivity between regions of the DAN increased with regions in the SalVentAttn network (e.g., anterior insula/IFG and anterior cingulate cortex). Second, areas within the PMC, part of the posterior core of the DMN, showed a pattern of manifold contraction that was primarily driven by a decrease in covariance with other DMN regions and an increase in covariance with regions of the superior DAN and also with several regions within the SalVentAttn network. Together, these results suggest that functional interactions of the sensorimotor system with the SalVentAttn network are important during initial learning. In other work, areas within the SalVentAttn network have been implicated in several aspects of cognitive control and motivation (*Botvinick et al., 2004*; *Holroyd and Yeung, 2012*; *Pearson et al., 2009*; *Shenhav et al., 2014*), and in studies on reward-based decision-making, these regions are thought to support exploratory behavior. For instance, several neuroimaging studies have shown that SalVentAttn areas are activated in response to novel or salient stimuli in the environment, presumably reflecting the engagement of attentional resources for sampling new information (*Corbetta et al., 2008*). In addition, recent work (*Badre et al., 2012*; *Blanchard and Gershman, 2018*; *Boorman et al., 2009*; *Daw et al., 2006*; *Hogeveen et al., 2022*; *Kolling et al., 2016*) implicates several SalVentAttn areas in information gathering functions, so as to optimize reward outcomes. This neural perspective is consistent with both others' (*Dam et al., 2013*) and our own behavioral findings (*de Brouwer et al., 2022*) that, during the early phases of reward-guided learning, performance is more variable, presumably as individuals explore the relationship between motor commands and associated sensory feedback.

During late learning, we observed that many of the changes in manifold architecture observed during early learning reversed. For instance, areas within both the DAN and DMN now exhibited

contraction along the manifold, whereas the PMC now exhibited expansion. Connectivity analyses showed that the contraction within the DAN and DMN was driven by increases in connectivity between these two networks—specifically, increases between premotor and superior parietal areas of the DAN with areas of the DMN. This suggests that once the mapping between motor commands and reward feedback have been learned, regions within the DAN and DMN become more integrated with one another. In the context of the current task, the shift in DAN connectivity from the SalVentAttn network during early learning to the DMN during late learning may reflect the hypothesized role of the DMN in supporting behavior using information from memory (*Buckner et al., 2008*; *Schacter et al., 2012*; *Smallwood et al., 2021*). This interpretation is consistent with prior work showing that connectivity between the DMN and premotor cortex tends to increase once rules have been learned (*Shamloo and Helie, 2016*), as well as studies from other task domains showing that the DMN contributes to behavior when actions must be guided by information from memory and a knowledge of task structure (*Murphy et al., 2019*; *Murphy et al., 2018*; *Vatansever et al., 2017*). Our analysis, therefore, provides additional evidence, albeit from the domain of human motor learning, that functional interactions between the DMN and brain regions involved in sensorimotor processes support a mode of action in which behavior must be guided by memory processes (*Smallwood et al., 2021*)—in this case, the history of reward information accrued across previous movement trajectories. Finally, it is important to note that the reversal pattern of effects noted above suggests that our findings during learning cannot be simply attributed to the introduction of reward feedback and/or the perturbation during early learning, as both of these task-related features are also present during late learning. In addition, these results cannot be simply explained due to the passage of time or increasing subject fatigue as this would predict a consistent directional change in eccentricity across the baseline, early, and Late learning epochs.

We also observed a relationship between changes in the manifold eccentricity of a subnetwork of the DAN (DAN-A) with subject learning performance. We found that the more this subnetwork contracted, and then subsequently expanded, along the manifold during early and late learning, respectively, the better subjects performed at the task. Our connectivity analysis revealed that this change in DAN-A activity was mainly driven by increases in connectivity with SalVentAttn subnetworks during early learning and by increases in connectivity with DMN subnetworks during late learning. Notably, areas of the DAN are not thought to generate top-down signals for response selection but instead transform the input signals they receive (e.g., related to reward, memory) into motor commands (*Luo et al., 2010*). Our analysis thus suggests that this system's contribution to motor behavior may be facilitated through changes in its functional coupling to both the SalVentAttn and DMN over time. To speculate, this shift in functional coupling may reflect a shift from more exploratory to more exploitative modes of behavior across early to late periods of motor learning, respectively. It is also possible that some of these task-related shifts in connectivity relate to shifts in task-general processes, such as changes in the allocation of attentional resources (*Bédard and Song, 2013*; *Rosenberg et al., 2016*) or overall cognitive engagement (*Aben et al., 2020*), which themselves play critical roles in shaping learning (*Codol et al., 2018*; *Holland et al., 2018*; *Song, 2019*; *Taylor and Thoroughman, 2008*; *Taylor and Thoroughman, 2007*; for a review of these topics, see *Tsay et al., 2023*). Such processes are particularly important during the earlier phases of learning when sensorimotor contingencies need to be established. While these remain questions for future work, our data nevertheless suggest that this shift in connectivity may be enabled through the PMC.

Although traditionally considered a member of the DMN (*Thomas Yeo et al., 2011*), studies have established that the PMC contains echoes of neural signals originating from across the cortex (*Leech et al., 2012*). In our study, this region initially became more segregated from the rest of the DMN and increased connectivity with the SalVentAttn network during early learning. However, during late learning, the PMC reduced its connectivity with the SalVentAttn network and became more integrated with other members of the DMN. This pattern of changes differed from other areas of the DMN, indicating that the PMC may serve a different function during motor learning than other areas of this system. Prior studies have linked the activity of PMC areas to reward processing (*Kable and Glimcher, 2007*; *Knutson and Bossaerts, 2007*; *McCoy and Platt, 2005*) and the selection of response strategies that attempt to optimize reward outcomes (*Barack et al., 2017*; *Pearson et al., 2009*; *Wan et al., 2015*). Consistent with this, recent research shows that areas within the PMC are able to integrate information over particularly long periods of time (*Hasson et al., 2015*; *Heilbronner and*

*Platt, 2013*; *Lerner et al., 2011*). This characteristic positions the PMC as an ideal candidate region to orchestrate the neural transition from (1) exploring the relationship between motor commands and sensory feedback during early learning to (2) subsequently exploiting this learned relationship during late learning. Taken together, these distinctive functional properties of PMC activity, coupled with its diverse patterns of whole-brain connectivity (*Hagmann et al., 2008*; *Hutchison et al., 2015*; *Margulies et al., 2009*), suggest an important role for this region in directing long-term behavioral adaptation in accordance with higher-order task objectives (*Braga et al., 2013*; *Pearson et al., 2011*).

While we identified several changes in the cortical manifold that are associated with reward-based motor learning, it is noteworthy that we did not observe any significant changes in manifold eccentricity within the striatum. While clearly the evidence indicates that this region plays a key role in reward-guided behavior (*Averbeck and O'Doherty, 2022*; *O'Doherty et al., 2017*), there are several possible reasons why our manifold approach did not identify this collection of brain areas. First, the relatively small size of the striatum may mean that our analysis approach was too coarse to identify changes in the connectivity of this region. Though we used a 3T scanner and employed a widely used parcellation scheme that divided the striatum into its constituent anatomical regions (e.g., hippocampus, caudate), both of these approaches may have obscured important differences in connectivity that exist *within* each of these regions. For example, areas such the hippocampus and caudate are not homogeneous areas but themselves exhibit gradients of connectivity (e.g., head versus tail) that can only be revealed at the voxel level (*Tian et al., 2020*; *Vos de Wael et al., 2021*). Second, while our dimension reduction approach, by design, aims to identify gradients of functional connectivity that account for the largest amounts of variance, the limited number of striatal regions (compared to cortex) necessitates that their contribution to the total whole-brain variance is relatively small. Consistent with this perspective, we found that the low-dimensional manifold architecture in cortex did not strongly depend on whether or not striatal regions were included in the analysis (see *Figure 3—figure supplement 2*). As such, selective changes in the patterns of functional connectivity at the level of the striatum may be obscured using our cortex × striatum dimension reduction approach. Future work can help address some of these limitations by using both finer parcellations of striatal cortex (perhaps even down to the voxel level) (*Tian et al., 2020*) and by focusing specifically on the changes in the interactions *between* the striatum and cortex during learning. The latter can be accomplished by selectively performing dimension reduction on the slice of the functional connectivity matrix that corresponds to functional coupling between striatum and cortex.

## Conclusions

Our study set out to characterize changes in the landscape of brain activity that underlies reward-based motor learning. We used dimensionality reduction techniques to build a manifold that describes changes in the functional organization of the cortex and striatum during different phases of learning. During early learning, we found that regions within the DAN and DMN became relatively segregated from each other, with the DAN becoming more integrated with the SalVentAttn network. This pattern reversed during later learning, with regions within the DAN now becoming more integrated with the DMN. Notably, regions of the PMC, within the posterior core of the DMN, showed the reverse pattern, exhibiting coupling with the SalVentAttn early during learning and with other regions of the DMN later during learning. Together, these findings provide a unique cortical perspective into the neural changes that underlie reward-based motor learning and point to marked transitions in the activity of transmodal cortical regions in organizing behavior over time.

## Materials and methods
### Participants

Forty-six right-handed individuals (27 females, aged 18–28 years) participated in the MRI study. Of these 46 participants, 10 individuals were removed from the final analysis either due to excessive head motion in the MRI scanner, incomplete scans, poor task compliance (i.e., >25% of trials not being completed within the maximal trial duration), or missing data (i.e., >20% of trials being missed due to insufficient pressure of the fingertip on the MRI-compatible tablet). We assessed right-handedness using the Edinburgh handedness questionnaire (*Oldfield, 1971*) and obtained informed consent before beginning the experimental protocol. The Queen's University Research Ethics Board approved

the study (ethics approval number: CNS-019-16), and it was conducted in coherence to the principles outlined in the Canadian Tri-Council Policy Statement on Ethical Conduct for Research Involving Humans and the principles of the Declaration of Helsinki (1964).

## Procedure

Prior to MRI testing, participants first took part in an MRI training session inside a mock (0T) scanner, made to look and sound like a real MRI scanner. This training session served multiple purposes. First, it introduced participants to the key features of the motor task that was subsequently performed in the MRI scanner. Second, it allowed us to screen for subjects who could obtain baseline performance levels on the task. Third, it allowed us to screen for subjects who could remain still for a long period of time without feeling claustrophobic. With respect to the latter, we monitored subjects' head movement in the mock scanner while they performed practice task trials and during simulated anatomical scans. This monitoring was done by attaching, via medical tape, a Polhemus sensor to each subject's forehead (Polhemus, Colchester, Vermont), which allowed a real-time read-out of subject head displacement in each of the three axes of translation and rotation (six dimensions total). Whenever subjects' head translation and/or rotation reached 0.5 mm or 0.5° rotation (within a prespecified velocity criterion), subjects received an unpleasant auditory tone, delivered through a speaker system located near the head. All subjects learned to constrain their head movement via this auditory feedback. Following this first training session, subjects who met our criteria were invited to subsequently participate in the reward-based motor learning task (see below for details), approximately 1 week later.

## Apparatus

During testing in the mock (0T) scanner, subjects performed hand movements that were directed toward a target by applying fingertip pressure on a digitizing touchscreen tablet (Wacom Intuos Pro M tablet). During the actual MRI testing sessions, subjects used an MRI-compatible digitizing tablet (Hybridmojo LLC, CA). In both the mock and real MRI scanner, the visual stimuli were rear-projected with an LCD projector (NEC LT265 DLP projector, 1024 × 768 resolution, 60 Hz refresh rate) onto a screen mounted behind the participant. The stimuli on the screen were viewed through a mirror fixated on the MRI coil directly above the participants' eyes, thus preventing the participant from being able to see their hand.

## Reward-based motor learning task

In the motor task, participants were trained, through reward-based feedback, to produce finger movement trajectories for an unseen shape. Specifically, subjects were instructed to repeatedly trace, without visual feedback of their actual finger paths, a subtly curved path displayed on the screen (the visible path, *Figure 1B and C*). Participants were told that, following each trial, they would receive a score based on how 'accurately' they traced the visible path. However, unbeknownst to them, they actually received points based on how well they traced the mirror-image path (the reward path, *Figure 1B and C*). Critically, because participants received no visual feedback about their actual finger trajectories or the 'rewarded' shape, they could not use error-based learning mechanisms to guide learning (*Pekny et al., 2015*; *Wolpert et al., 2011*). This task was a modification on the motor tasks developed by *Dam et al., 2013*; *Wu et al., 2014*.

Each trial started with the participant moving a cursor (3 mm radius cyan circle), which represented their finger position, into the start position (4 mm radius white circle) at the bottom of the screen (by sliding the index finger on the tablet). The cursor was only visible when it was within 30 mm of the start position. After the cursor was held within the start position for 0.5 s, the cursor disappeared and a rightward-curved path (visible path) and a movement distance marker appeared on the screen (*Figure 1B*). The movement distance marker was a horizontal red line (30 × 1 mm) that appeared 60 mm above the start position. The visible path connected the start position and movement distance marker, and had the shape of a half sine wave with an amplitude of 0.15 times the marker distance. Participants were instructed to trace the curved path. When the cursor reached the target distance, the target changed color from red to green to indicate that the trial was completed. Importantly, other than this color change in the distance marker, the visible curved path remained constant and participants never received any feedback about the position of their cursor.

In the baseline block, participants did not receive any feedback about their performance. In the learning block, participants were rewarded 0–100 points after reaching the movement distance marker, and were instructed to do their best to maximize this score across trials (following the movement, the points were displayed as text centrally on the screen). Each trial was terminated after 4.5 s, independent of whether the cursor had reached the target. After a delay of 1.5 s (during which the screen was blanked), allowing time to save the data and the subject to return to the starting location, the next trial started with the presentation of the start position.

To calculate the reward score on each trial in the learning block, the x position of the cursor was interpolated at each cm displacement from the start position in the y direction (i.e., at exactly 10, 20, 30, 40, 50, and 60 mm). For each of the six y positions, the absolute distance between the interpolated x position of the cursor and the x position of the rewarded path was calculated. The sum of these errors was scaled by dividing it by the sum of errors obtained for a half cycle sine-shaped path with an amplitude of 0.5 times the target distance, and then multiplied by 100 to obtain a score ranging between 0 and 100. The scaling worked out such that a perfectly traced visible path would result in an imperfect score of 40 points. This scaling was chosen on the basis of extensive pilot testing in order to (1) encourage motor exploration across trials (in search of higher scores), (2) achieve variation across subjects in overall performance (i.e., individual differences in learning curves), and (3) ensure that subjects still received informative score feedback when tracing in the vicinity of the visible trajectory.

During the training session in the mock MRI scanner (i.e., ~1 week prior to the MRI testing session), participants performed only a practice block in which they traced a straight line, first with (40 trials) and then without (40 trials), visual feedback of the position of the cursor during the movement (80 trials total). This training session exposed participants to several key features of the task (e.g., use of the touchscreen tablet, trial timing, removal of cursor feedback) and allowed us to establish adequate performance levels. Importantly, however, subjects did not encounter any reward-based feedback (reward scores) during this initial training session.

At the beginning of the MRI testing session, but prior to the first scan being collected, participants re-acquainted themselves with the motor task by first performing a practice block in which they traced a straight line with (40 trials) and then without (40 trials) visual feedback of the position of the cursor. Next, we collected an anatomical scan, followed by a DTI scan, followed by a resting-state fMRI scan. During the latter resting-state scan, participants were instructed to rest with their eyes open while fixating on a central cross location presented on the screen. (Note that the DTI and resting-state fMRI data are not the focus on the present study.) Next, participants performed the motor task, which consisted of two separate experimental runs without visual feedback of the cursor: (1) a baseline block of 70 trials in which they attempted to trace the curved path and no score feedback was provided, and (2) a separate learning block of 200 trials in which participants were instructed to maximize their score shown at the end of each trial. Note that, at the end of testing, we did not assess participants' awareness of the manipulation (i.e., that they were, in fact, being rewarded based on a mirror image path of the visible path). This experiment was only performed once.

## Behavioral data analysis

### Data preprocessing

Each movement trajectory was first re-sampled to 10 equally spaced points along the y (vertical) axis, between the starting position and the target distance marker. We defined subjects' reaction time (RT) as the time between trial onset and the cursor reaching 10% of the distance from the starting location, and defined subjects' movement time (MT) as the remaining time until reaching the target distance marker. To quantify the variability of the drawn path as a function of trial number, we used the following method. First, we calculated the average path at each trial by applying a moving average smoother (window size seven trials) to the sequence of paths drawn by each participant. Then, for each trial, path variability was measured by the mean absolute x position difference between the actual path drawn and the average path in that trial (across the 10 sample points equally spaced along the y direction). Specifically, path variability was quantified in the following manner:

Path trial $t$: $X_t^k$ ($k = 1, 2, \ldots, 10$, i.e., 10 sample points)

For each $t$, $\overline{X}_t^k = \frac{1}{7} \sum_{i=t-3}^{t+3} X_i^k$ (sliding window mean, half window width 3)

Variability of path in trial $t$: $\frac{1}{10} \sum_{k=1}^{10} \left| X_t^k - \overline{X}_t^k \right|$

Trials in which the cursor did not reach the target within the time limit were excluded from the offline analysis of hand movements (~1% of trials). As insufficient pressure on the touchpad resulted in a default state in which the cursor was reported as lying in the top-left corner of the screen, we excluded trials in which the cursor jumped to this position before reaching the target region (~2% of trials). We then applied a conservative threshold on the MT and RT, removing the top 0.05% of trials across all subjects. As the motor task did not involve response discrimination, we did not set a lower threshold on these variables.

## Functional PCA of subject behavioral data

All subject behavioral data were averaged over eight trial bins. We represented individual learning curves as functional data using a cubic spline basis with smoothing penalty estimated by generalized cross-validation (*Härdle, 1990*). We then performed *functional PCA* (*Ramsay and Silverman, 2013*), which allowed us to extract components capturing the dominant patterns of variability in subject performance. Using this analysis, we found that the top component, which describes overall learning, explained a majority of the variability (~75%) in performance. Spline smoothing and fPCA were performed using the R package fda (*Ramsay et al., 2022*).

## MRI acquisition

Participants were scanned using a 3-Tesla Siemens TIM MAGNETOM Trio MRI scanner located at the Centre for Neuroscience Studies, Queen's University (Kingston, ON, Canada). Subject anatomicals were acquired using a 32-channel head coil and a T1-weighted ADNI MPRAGE sequence (TR = 1760 ms, TE = 2.98 ms, field of view = 192 mm × 240 mm × 256 mm, matrix size = 192 × 240 × 256, flip angle = 9°, 1 mm isotropic voxels). This was followed by a series of diffusion-weighted scans and a resting-state scan (which are not the focus of the present investigation). Next, we acquired fMRI volumes using a T2*-weighted single-shot gradient-echo echo-planar imaging (EPI) acquisition sequence (time to repetition [TR] = 2000 ms, slice thickness = 4 mm, in-plane resolution = 3 mm × 3 mm, time to echo [TE] = 30 ms, field of view = 240 mm × 240 mm, matrix size = 80 × 80, flip angle = 90°, and acceleration factor [integrated parallel acquisition technologies, iPAT] = 2 with generalized auto-calibrating partially parallel acquisitions [GRAPPA] reconstruction). Each volume comprised 34 contiguous (no gap) oblique slices acquired at an ~30° caudal tilt with respect to the plane of the anterior and posterior commissure (AC-PC), providing whole-brain coverage of the cerebrum and cerebellum. Note that for the current study we did not examine the changes in cerebellar activity during learning. For the baseline and learning scans, we acquired 222 and 612 imaging volumes, respectively. Each of these task-related scans included an additional six imaging volumes at both the beginning and end of the scan. On average, the total MRI scanning session lasted ~2 hrs (including setup time and image acquisition).

## fMRI preprocessing

Preprocessing of anatomical and functional MRI data was performed using fMRIPrep 20.1.1 (*Esteban et al., 2019*, *Esteban et al., 2024*; RRID:SCR_016216) which is based on Nipype 1.5.0 (*Gorgolewski et al., 2011*; *Gorgolewski et al., 2018*; RRID:SCR_002502). Many internal operations of fMRIPrep use Nilearn 0.6.2 (*Abraham et al., 2014*; RRID:SCR_001362), mostly within the functional processing workflow. For more details of the pipeline, see the section corresponding to workflows in fMRIPrep's documentation. Below we provide a condensed description of the preprocessing steps.

T1w images were corrected for intensity non-uniformity (INU) with N4BiasFieldCorrection (*Tustison et al., 2010*), distributed with ANTs 2.2.0 (*Avants et al., 2008*; RRID:SCR_004757). The T1w-reference was then skull-stripped with a Nipype implementation of the antsBrainExtraction.sh workflow (from ANTs), using OASIS30ANTs as target template. Brain tissue segmentation of cerebrospinal fluid (CSF), white matter (WM), and gray matter (GM) was performed on the brain-extracted T1w using fast (FSL 5.0.9, RRID:SCR_002823)(*Zhang et al., 2001*). A T1w-reference map was computed after registration of the T1w images (after INU-correction) using mri_robust_template (FreeSurfer 6.0.1; *Reuter et al., 2010*). Brain surfaces were reconstructed using recon-all (FreeSurfer 6.0.1, RRID:SCR_001847; *Dale et al., 1999*), and the brain mask estimated previously was refined with a custom variation of the method to reconcile ANTs-derived and FreeSurfer-derived segmentations of the cortical gray-matter of Mindboggle (RRID:SCR_002438; *Klein et al., 2017*). Volume-based spatial normalization to

standard space (MNI152NLin6Asym) was performed through nonlinear registration with antsRegistration (ANTs 2.2.0), using brain-extracted versions of both T1w reference and the T1w template.

For each BOLD run, the following preprocessing was performed. First, a reference volume and its skull-stripped version were generated using a custom methodology of fMRIPrep. Head-motion parameters with respect to the BOLD reference (transformation matrices, and six corresponding rotation and translation parameters) are estimated before any spatiotemporal filtering using mcflirt (FSL 5.0.9; *Jenkinson et al., 2002*). BOLD runs were slice-time corrected using 3dTshift from AFNI 20160207 (*Cox and Hyde, 1997*; RRID:SCR_005927). The BOLD reference was then co-registered to the T1w reference using bbregister (FreeSurfer) which implements boundary-based registration (*Greve and Fischl, 2009*). Co-registration was configured with six degrees of freedom. The BOLD time series were resampled with a single interpolation step by composing all the pertinent transformations (i.e. head-motion transform matrices, and co-registrations to anatomical and output spaces). BOLD time series were resampled onto their original, native space, as well as standard space (MNI152NLin6Asym), using antsApplyTransforms (ANTs), configured with Lanczos interpolation to minimize the smoothing effects of other kernels (*Lanczos, 1964*). Striatal data in standard space was combined with resampled BOLD time series on the fsaverage surface to produce Grayordinates files (*Glasser et al., 2013*) containing 91k samples, using fsaverage as the intermediate standardized surface space. Resampling onto fsaverage was performed using mri_vol2surf (FreeSurfer).

A set of 34 motion and physiological regressors were extracted in order to mitigate the impact of head motion and physiological noise. The six head-motion estimates calculated in the correction step were expanded to include temporal derivatives and quadratic terms of each of the original and derivative regressors, totaling 24 head-motion parameters (*Satterthwaite et al., 2013*). Ten component-based physiological regressors were estimated using the aCompCor approach (*Behzadi et al., 2007*; *Muschelli et al., 2014*), where the top five PCs were separately extracted from WM and CSF masks. PCs were estimated after high-pass filtering the preprocessed BOLD time series (using a discrete cosine filter with 128 s cut-off).

### Regional time-series extraction

For each participant and scan, the average BOLD time series were computed from the grayordinate time series for (1) each of the 998 regions defined according to the Schaefer 1000 parcellation (*Schaefer et al., 2018*; two regions are removed from the parcellation due to their small parcel size) and (2) each of the 12 striatal regions defined according to the Harvard-Oxford atlas (*Frazier et al., 2005*; *Makris et al., 2006*), which included the caudate, putamen, accumbens, pallidum, hippocampus, and amygdala. Region time series were denoised using the above-mentioned confound regressors in conjunction with the discrete cosine regressors (128 s cut-off for high-pass filtering) produced from fMRIprep and low-pass filtering using a Butterworth filter (100 s cut-off) implemented in Nilearn. Finally, all region time series were z-scored.

## Neuroimaging data analysis

### Covariance estimation and centering

For every participant, region time series from the task scans were spliced into three equal-lengthed task epochs (210 imaging volumes each), after having discarded the first six imaging volumes (thus avoiding scanner equilibrium effects). This allowed us to estimate functional connectivity from continuous brain activity over the corresponding 70 trials for each epoch; Baseline comprised of the initial 70 trials in which subjects performed the motor task in the absence of any reward feedback, whereas the early and late learning epochs consisted of the first and last 70 trials after the onset of reward feedback, respectively. Then, we separately estimated functional connectivity matrices for each epoch by computing the region-wise covariance matrices using the Ledoit–Wolf estimator (*Ledoit and Wolf, 2004*). Note that our use of equal-length epochs for the three phases ensured that no biases in covariance estimation were introduced due to differences in time-series length.

Next, we centered the connectivity matrices using the approach advocated by *Zhao et al., 2018*, which leverages the natural geometry of the space of covariance matrices (*Areshenkoff et al., 2022*; *Areshenkoff et al., 2021*). In brief, this involved adjusting the covariance matrices of each participant to have a common mean, which was equivalent to the overall mean covariance, thus removing subject-specific variations in functional connectivity. First, a grand mean covariance matrix, $S_{gm}$, was

computed by taking the geometric mean covariance matrix across all $i$ participants and $j$ epochs. Then, for each participant we computed the geometric mean covariance matrix across task epochs, $\bar{S}_i$ , and each task epoch covariance matrix $S_{ij}$ was projected onto the tangent space at this mean participant covariance matrix $S_i$ to obtain a tangent vector

$$T_{ij} = \bar{S}_i^{\,1/2} \, \log \left( \bar{S}_i^{\,-1/2} \, S_{ij} \bar{S}_i^{\,-1/2} \right) \bar{S}_i^{\,1/2}$$

where log denotes the matrix logarithm. We then transported each tangent vector to the grand mean $\bar{S}_{gm}$ using the transport proposed by *Zhao et al., 2018*, obtaining a centered tangent vector

$$T_{ij}^c = GT_{ij}G^\top$$

where $G = \bar{S}_{gm}^{\,1/2}S_i^{-1/2}$ . This centered tangent vector now encodes the same difference in covariance, but now expressed relative to the grand mean. Finally, we projected each centered tangent vector back onto the space of covariance matrices, to obtain the centered covariance matrix

$$S_{ij}^c = \bar{S}_{gm}^{\,1/2} \exp \left( \bar{S}_{gm}^{\,-1/2} \, T_{ij}^c \bar{S}_{gm}^{\,-1/2} \right) \bar{S}_{gm}^{\,1/2}$$

where $exp$ denotes the matrix exponential. For the benefits of this centering approach, see *Figure 2*, and for an additional overview, see *Areshenkoff et al., 2022*.

Note that we have implemented many of the computations required to replicate the analysis in an publicly available R package **spdm**, which is freely available from GitHub (*Areshenkoff, 2023*).

## Manifold construction

Connectivity manifolds for all centered functional connectivity matrices were derived in the following steps. First, consistent with previous studies (*Gale et al., 2022*; *Hong et al., 2020*; *Margulies et al., 2016*; *Vos de Wael et al., 2020*), we applied row-wise thresholding to retain the top 10% connections in each row, and then computed cosine similarity between each row to produce an affinity matrix that describes the similarity of each region's connectivity profiles. Second, we applied PCA to obtain a set of PCs that provide a low-dimensional representation of connectivity structure (i.e., connectivity gradients). We selected PCA as our dimension reduction technique based on recent work demonstrating the improved reliability of PCA over nonlinear dimensionality reduction techniques (e.g., diffusion map embedding; *Hong et al., 2020*).

To provide a basis for comparing changes in functional network architecture that arise during learning specifically, we constructed a template manifold using the same aforementioned manifold construction procedures from a group-average baseline connectivity matrix that was derived from the geometric mean (across participants) of all centered baseline connectivity matrices. We aligned all individual manifolds (36 participants × 3 epochs; 108 total) to this baseline template manifold using Procrustes alignment. All analyses on the aligned manifolds were performed using the top three PCs, which cumulatively explained ~70% of the total variance in the template manifold. Across participants and epochs, the top three PCs, following Procrustes alignment, had an average correlation of $r = 0.88$ with their respective PCs in the template manifold, thus demonstrating good overall reliability and alignment across participants and epochs. Together, this approach enabled us to uniquely examine the learning-related changes in manifold structure (during early and late learning), and specifically how these deviate from the baseline task functional architecture.

## Manifold eccentricity and analyses

Recent work has quantified the embedding of regions and networks in low-dimensional spaces using Euclidean distance as a measure (*Bethlehem et al., 2020*; *Park et al., 2021a*; *Park et al., 2021b*; *Valk et al., 2023*). 'Eccentricity' refers to the Euclidean distance between a single region and the manifold centroid (*Park et al., 2021a*), which, in the case of PCA, is equivalent to a region's magnitude, or vector length. Note that eccentricity provides a scalar index of network integration and segregation, in which distal regions with greater eccentricity are more segregated than proximal regions

that integrate more broadly across functional networks (*Park et al., 2021a*; *Park et al., 2021b*; *Valk et al., 2023*). To validate this interpretation with our own data, we correlated the baseline template manifold eccentricity with three graph theoretical measures of functional integration and segregation. These measures were calculated on the row-wise thresholded template connectivity matrix and included *node strength*, which is the sum of a region's connectivity weights; *within-module degree z-score*, which measures the degree centrality of a region within its respective network; and *participation coefficient*, which measures the network diversity of a region's connectivity distribution (*Rubinov and Sporns, 2010*). Regions were assigned to their respective intrinsic functional networks (*Schaefer et al., 2018*; *Thomas Yeo et al., 2011*) for calculations of within-module degree z-score and participation coefficient.

We computed eccentricity for each brain region for all individual manifolds (each participant and epoch). This allowed us to observe manifold expansions (increases in eccentricity) and contractions (decreases in eccentricity) throughout early and late learning, thereby probing learning-related changes in functional segregation and integration (e.g., see *Figure 4*). We compared region eccentricity between the baseline, early, and late learning epochs by performing a series of region-wise paired *t*-tests between these three key epochs. We applied FDR correction (q < 0.05) across regions for each contrast.

## Seed connectivity analyses

In order to understand the underlying changes in regional covariance that ultimately give rise to the observed changes in manifold eccentricity, we performed seed connectivity contrasts between the different task epochs. To this end, we selected several seed regions that were statistically significant in the early learning > baseline contrast, which included areas in the left MPFC, left premotor cortex (PMd), and left PMC, thereby allowing us to characterize some of the cortical and striatal changes that are associated reward-based learning. For completeness, we also selected homologous regions in the right hemisphere (see *Figure 5—figure supplement 1*). For each seed region, we generated functional connectivity maps for the epochs of interest in every participant and computed region-wise paired *t*-tests for both the early > baseline and late > early contrasts. For all contrasts, we opted to show the unthresholded *t*-maps so as to visualize the complete multivariate pattern of connectivity changes that drive changes in eccentricity (a multivariate measure). In addition, we constructed spider plots further characterizing these changes at the network level by averaging the *t*-values across individual regions according to their network assignment (*Thomas Yeo et al., 2011*). Note that these analyses are mainly intended to provide characterization (and interpretation) of the connectivity changes of representative regions from our main eccentricity analyses.

## Behavioral correlation analyses

To investigate the relationship between changes in manifold structure and individual differences in learning performance, we computed a correlation, across participants, between learning score and each region's change in eccentricity from baseline to early learning (*Figure 6C*) and from early to late learning (*Figure 6D*). This produced two correlation maps, one for each contrast. We found that the spatial specificity of significant regions in these correlation maps overlapped substantially with the dorsal attention A (DAN-A) network, from the 17-network Schaefer 1000 assignments (*Schaefer et al., 2018*). This was determined by evaluating the mean correlation (across regions) for each of the 17-network assignments against a null distribution generated by projecting each brain region's correlation onto their respective Schaefer 1000 parcels on the 32k fsLR spherical mesh and performing 1000 iterations of the Váša spin-testing permutation procedure (*Markello et al., 2022*; *Váša et al., 2018*). This allowed us to generate, for each brain network and pairwise comparison (baseline to early learning and early to late learning), a topographical distribution of correlations that could be expected simply due to chance from spatial autocorrelations in the brain maps (see *Figure 6E and F*). We then empirically assessed the statistical significance of our real correlation values against this spatial null distribution for each brain network. Because the spin-testing procedure assesses only the probability of having observed a given correlation value, and not whether that correlation value itself differs from zero, we incorporated the additional stipulation that an effect would be deemed significant only if the real correlation value was also statistically different from zero (at p<0.05).

To explore the underlying changes in functional connectivity that give way to these brain–behavior correlations, we performed seed connectivity contrasts, using paired *t*-tests on the 17-network parcellation, between the different task epochs (early > baseline and late > early). As with the 'Seed connectivity analyses' section above, we opted to show the unthresholded *t*-maps so as to visualize the complete multivariate pattern of connectivity that underlies the brain–behavior correlations (*Figure 6G*). Together, these complementary approaches enabled us to explore how individual differences in performance relate to changes in manifold structure at the region- and network levels.

## Acknowledgements

This work was supported by operating grants from the Canadian Institutes of Health Research Grant (MOP126158), the Natural Sciences and Engineering Research Council (RGPIN-2017-04684), and Botterell Foundation Award, as well as funding from the Canadian Foundation for Innovation (35559). The authors would like to thank Martin York and Sean Hickman for technical assistance, and Don O'Brien for assistance with data collection.

## Additional information

### Competing interests

Daniel J Gale, Jason Gallivan: is an employee of Voxel AI Inc. This funding source had no role in the design, management, data analysis, presentation, or interpretation and write-up of the data. The other authors declare that no competing interests exist.

### Funding

| Funder | Grant reference number | Author |
| --- | --- | --- |
| Canadian Institutes of Health Research | PJT175012 | Jason Gallivan |
| Natural Sciences and Engineering Research Council of Canada | RGPIN-2017-04684 | Jason Gallivan |

The funders had no role in study design, data collection and interpretation, or the decision to submit the work for publication.

### Author contributions

Qasem Nick, Resources, Software, Formal analysis, Validation, Investigation, Visualization, Writing – original draft, Writing – review and editing; Daniel J Gale, Resources, Software; Corson Areshenkoff, Resources, Investigation, Methodology, Formal analysis, Software; Anouk De Brouwer, Formal analysis, Investigation, Methodology, Resources; Joseph Nashed, Investigation, Methodology; Jeffrey Wammes, Resources; Tianyao Zhu, Resources, Formal analysis; Randy Flanagan, Conceptualization, Resources, Writing – review and editing; Jonny Smallwood, Visualization, Writing – original draft, Writing – review and editing; Jason Gallivan, Conceptualization, Resources, Supervision, Funding acquisition, Investigation, Writing – original draft, Project administration, Writing – review and editing, Methodology

### Author ORCIDs

Jeffrey Wammes ⓘ https://orcid.org/0000-0002-8923-5441
Jason Gallivan ⓘ https://orcid.org/0000-0002-7362-109X

### Ethics

Informed consent, and consent to publish, was obtained from the human participants.The Queen's University Research Ethics Board approved the study (ethics approval number: CNS-019-16) and it was conducted in coherence to the principles outlined in the Canadian Tri-Council Policy Statement on Ethical Conduct for Research Involving Humans and the principles of the Declaration of Helsinki (1964).

Reviewer #1 (Public Review): https://doi.org/10.7554/eLife.91928.3.sa1
Reviewer #2 (Public Review): https://doi.org/10.7554/eLife.91928.3.sa2
Reviewer #3 (Public Review): https://doi.org/10.7554/eLife.91928.3.sa3
Author response https://doi.org/10.7554/eLife.91928.3.sa4

## Additional files

### Supplementary files
• MDAR checklist

### Data availability

Versions of the data used specifically for analysis and the generation of the manuscript figures can be found on Dryad. In addition, BIDS-formatted versions of the raw data can be obtained at OpenNeuro: https://openneuro.org/datasets/ds005230/versions/1.0.0. Imaging data were preprocessed using fmriPrep, which is open source and freely available. Operations on covariance matrices, including estimation and centering, were performed using the R package spdm, which is freely available on GitHub (*Areshenkoff, 2023*). The analysis code for the paper can be found on GitHub (copy archived at *Nick, 2024*).

The following datasets were generated:

| Author(s) | Year | Dataset title | Dataset URL | Database and Identifier |
| --- | --- | --- | --- | --- |
| Nick Q, Gale DJ, Areshenkoff CN, De Brouwer AJ, Nashed JY, Wammes J, Flanagan JR, Smallwood J, Gallivan JP | 2024 | Reconfigurations of cortical manifold structure during reward-based motor learning | https://doi.org/10.5061/dryad.7sqv9s512 | Dryad Digital Repository, 10.5061/dryad.7sqv9s512 |
| Nick Q, Gale DJ, Areshenkoff CN, De Brouwer AJ, Nashed JY, Wammes J, Flanagan JR, Smallwood J, Gallivan JP | 2024 | Reinforcement-Learning Generalization | https://doi.org/10.18112/openneuro.ds005230.v1.0.0 | OpenNeuro, 10.18112/openneuro.ds005230.v1.0.0 |

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
