## [Editor Report · eLife assessment]

This **valuable** study uses **convincing** state-of-the-art neuroimaging analyses to characterize whole-brain networks during reward-based motor learning. This work motivates future research to dissociate why the observed changes in neural connectivity occur and how they support reward-based motor learning. The study is highly relevant for researchers at the intersection of decision-making and sensorimotor learning.

---

## [Referee Report · Reviewer #1 (Public Review)]

This important study uses a wide variety of convincing, state-of-the-art neuroimaging analyses to characterize whole-brain networks and relate them to reward-based motor learning. During early learning, the authors found increased covariance between the sensorimotor and dorsal attention networks, coupled with reduced covariance between the sensorimotor and default mode networks. During late learning, they observed the opposite pattern. It remains to be seen whether these changes reflect generic changes in task engagement during learning or are specific to reward-based motor learning. This study is highly relevant for researchers interested in reward-based motor learning and decision-making.

---

## [Referee Report · Reviewer #2 (Public Review)]

This useful investigation of learning-driven dynamics of cortical and some subcortical structures combines a novel in-scanner learning paradigm with interesting analysis approaches. The new task for reward-based motor learning is compelling and goes beyond the current state of the art. The results are of interest to neuroscientists working on motor control and reward-based learning.

Comments on revised version:

The revision has produced a stronger manuscript. Thank you for your thorough responses to the comments and concerns.

---

## [Referee Report · Reviewer #3 (Public Review)]

Summary:

The manuscript of Nick and colleagues addresses the intriguing question of how brain connectivity evolves during reward-based motor learning. The concept of quantifying connectivity through changes in extraction and contraction across lower-dimensional manifolds is both novel and interesting and the presented results are clear and well-presented. Overall, the manuscript is a valuable addition to the field.

Strengths:

This manuscript is written in a clear and comprehensible way. It introduces a rather novel technique of assessing connectivity across lower-dimensional manifold which has hitherto not been applied in this way to the question of reward-based motor learning. Thus, this presents a unique viewpoint on understanding how the brain changes with motor learning. I particularly enjoyed the combination of connectivity-based, followed by further scrutiny of seed-based connectivity analyses, thus providing a more comprehensive viewpoint. Now it also has added a more comprehensive report on the behavioural changes of learning, and the added statistical quantification, which is useful.

Weaknesses:

The main weakness of the manuscript is the lack of direct connection between the reported neural changes and behavioural learning. Namely, most of the results could also be explained by changes in attention allocation during the session, or changes in movement speed (independent of learning). The authors acknowledge some of these potential confounds and argue that factors like attention are important for learning. While this is true, it is nonetheless very limiting if one cannot ascertain whether the observed effects are due to attention (independent of learning) or attention allocated in the pursuit of learning. The only direct analysis linking behavioural changes to neural changes is based on individual differences in learning performance, where the DAN-A shows the opposite trend than group level effects, which they interpret as differences given the used higher-resolution parcellation. However, it could be that these learning effects are indeed much smaller and subtler compared to more dominant group-level attention effects during the task. The lack of a control condition in the task limits the interpretability of results as learning-related.

---

## [Author Response]

The following is the authors’ response to the original reviews.

**Recommendations**
Recommendation #1: Address potential confounds in the experimental design:(1a) Confounding factors between baseline to early learning. While the visual display of the curved line remains constant, there are at least three changes between these two phases: (1) the presence of reward feedback (the focus of the paper); (2) a perturbation introduced to draw a hidden, mirror-symmetric curved line; (3) instructions provided to use reward feedback to trace the line on the screen (intentionally deceitful). As such, it remains unclear which of these factors are driving the changes in both behavior and bold signals between the two phases. The absence of a veridical feedback phase in which participants received reward feedback associated with the shown trajectory seems like a major limitation.(1b) Confounding Factors Between Early and Late Learning. While the authors have focused on interpreting changes from early to late due to the explore-exploit trade-off, there are three additional factors possibly at play: (1) increasing fatigue, (2) withdrawal of attention, specifically related to individuals who have either successfully learned the perturbation within the first few trials or those who have simply given up, or (3) increasing awareness of the perturbation (not clear if subjective reports about perturbation awareness were measured.). I understand that fMRI research is resource-intensive; however, it is not clear how to rule out these alternatives with their existing data without additional control groups. [Another reviewer added the following: Why did the authors not acquire data during a control condition? How can we be confident that the neural dynamics observed are not due to the simple passage of time? Or if these effects are due to the task, what drives them? The reward component, the movement execution, increased automaticity?]

We have opted to address both of these points above within a single reply, as together they suggest potential confounding factors across the three phases of the task. We would agree that, if the results of our pairwise comparisons (e.g., Early > Baseline or Late > Early) were considered in isolation from one another, then these critiques of the study would be problematic. However, when considering the pattern of effects across the three task phases, we believe most of these critiques can be dismissed. Below, we first describe our results in this context, and then discuss how they address the reviewers’ various critiques.

Recall that from Baseline to Early learning, we observe an expansion of several cortical areas (e.g., core regions in the DMN) along the manifold (red areas in Fig. 4A, see manifold shifts in Fig. 4C) that subsequently exhibit contraction during Early to Late learning (blue areas in Fig. 4B, see manifold shifts in Fig. 4D). We show this overlap in brain areas in Author response image 1 below, panel A. Notably, several of these brain areas appear to contract back to their original, Baseline locations along the manifold during Late learning (compare Fig. 4C and D). This is evidenced by the fact that many of these same regions (e.g., DMN regions, in Author response image 1 panel A below) fail to show a significant difference between the Baseline and Late learning epochs (see Author response image 1 panel B below, which is taken from supplementary Fig 6). That is, the regions that show significant expansion and subsequent contraction (in Author response image 1 panel A below) tend not to overlap with the regions that significantly changed over the time course of the task (in Author response image 1 panel B below).

**Author response image 1. sa4fig1:** 

Note that this basic observation above is not only true of our regional manifold eccentricity data, but also in the underlying functional connectivity data associated with individual brain regions. To make this second point clearer, we have modified and annotated our Fig. 5 and included it below. Note the reversal in seed-based functional connectivity from Baseline to Early learning (leftmost brain plots) compared to Early to Late learning (rightmost brain plots). That is, it is generally the case that for each seed-region (A-C) the areas that increase in seed-connectivity with the seed region (in red; leftmost plot) are also the areas that decrease in seed-connectivity with the seed region (in blue; rightmost plot), and vice versa. [Also note that these connectivity reversals are conveyed through the eccentricity data — the horizontal red line in the rightmost plots denote the mean eccentricity of these brain regions during the Baseline phase, helping to highlight the fact that the eccentricity of the Late learning phase reverses back towards this Baseline level].

**Author response image 2. sa4fig2:** 

Critically, these reversals in brain connectivity noted above directly counter several of the critiques noted by the reviewers. For instance, this reversal pattern of effects argues against the idea that our results during Early Learning can be simply explained due to the (i) presence of reward feedback, (ii) presence of the perturbation or (iii) instructions to use reward feedback to trace the path on the screen. Indeed, all of these factors are also present during Late learning, and yet many of the patterns of brain activity during this time period revert back to the Baseline patterns of connectivity, where these factors are absent. Similarly, this reversal pattern strongly refutes the idea that the effects are simply due to the passage of time, increasing fatigue, or general awareness of the perturbation. Indeed, if any of these factors alone could explain the data, then we would have expected a gradual increase (or decrease) in eccentricity and connectivity from Baseline to Early to Late learning, which we do not observe. We believe these are all important points when interpreting the data, but which we failed to mention in our original manuscript when discussing our findings.

We have now rectified this in the revised paper, where we now write in our Discussion:

“Finally, it is important to note that the reversal pattern of effects noted above suggests that our findings during learning cannot be simply attributed to the introduction of reward feedback and/or the perturbation during Early learning, as both of these task-related features are also present during Late learning. In addition, these results cannot be simply explained due to the passage of time or increasing subject fatigue, as this would predict a consistent directional change in eccentricity across the Baseline, Early and Late learning epochs.”

However, having said the above, we acknowledge that one potential factor that our findings cannot exclude is that they are (at least partially) attributable to changes in subjects’ state of attention throughout the task. Indeed, one can certainly argue that Baseline trials in our study don’t require a great deal of attention (after all, subjects are simply tracing a curved path presented on the screen). Likewise, for subjects that have learned the hidden shape, the Late learning trials are also likely to require limited attentional resources (indeed, many subjects at this point are simply producing the same shape trial after trial). Consequently, the large shift in brain connectivity that we observe from Baseline to Early Learning, and the subsequent reversion back to Baseline-levels of connectivity during Late learning, could actually reflect a heightened allocation of attention as subjects are attempting to learn the (hidden) rewarded shape. However, we do not believe that this would reflect a ‘confound’ of our study per se — indeed, any subject who has participated in a motor learning study would agree that the early learning phase of a task is far more cognitively demanding than Baseline trials and Late learning trials. As such, it is difficult to disentangle this ‘attention’ factor from the learning process itself (and in fact, it is likely central to it).

Of course, one could have designed a ‘control’ task in which subjects must direct their attention to something other than the learning task itself (e.g., divided attention paradigm, e.g., Taylor & Thoroughman, 2007, 2008), and/or perform a secondary task concurrently (Codol et al., 2018; Holland et al., 2018), but we know that this type of manipulation impairs the learning process itself. Thus, in such a case, it wouldn’t be obvious to the experimenter what they are actually measuring in brain activity during such a task. And, to extend this argument even further, it is true that any sort of brain-based modulation can be argued to reflect some ‘attentional’ process, rather than modulations related to the specific task-based process under consideration (in our case, motor learning). In this regard, we are sympathetic to the views of Richard Andersen and colleagues who have eloquently stated that “The study of how attention interacts with other neural processing systems is a most important endeavor. However, we think that over-generalizing attention to encompass a large variety of different neural processes weakens the concept and undercuts the ability to develop a robust understanding of other cognitive functions.” (Andersen & Cui, 2007, Neuron). In short, it appears that different fields/researchers have alternate views on the usefulness of attention as an explanatory construct (see also articles from Hommel et al., 2019, “No one knows what attention is”, and Wu, 2023, “We know what attention is!”), and we personally don’t have a dog in this fight. We only highlight these issues to draw attention (no pun intended) that it is not trivial to separate these different neural processes during a motor learning study.

Nevertheless, we do believe these are important points worth flagging for the reader in our paper, as they might have similar questions. To this end, we have now included in our Discussion section the following text:

“It is also possible that some of these task-related shifts in connectivity relate to shifts in task-general processes, such as changes in the allocation of attentional resources (Bédard and Song, 2013; Rosenberg et al., 2016) or overall cognitive engagement (Aben et al., 2020), which themselves play critical roles in shaping learning (Codol et al., 2018; Holland et al., 2018; Song, 2019; Taylor and Thoroughman, 2008, 2007; for a review of these topics, see Tsay et al., 2023). Such processes are particularly important during the earlier phases of learning when sensorimotor contingencies need to be established. While these remain questions for future work, our data nevertheless suggest that this shift in connectivity may be enabled through the PMC.”

Finally, we should note that, at the end of testing, we did not assess participants' awareness of the manipulation (i.e., that they were, in fact, being rewarded based on a mirror image path). In hindsight, this would have been a good idea and provided some value to the current project. Nevertheless, it seems clear that, based on several of the learning profiles observed (e.g., subjects who exhibited very rapid learning during the Early Learning phase, more on this below), that many individuals became aware of a shape approximating the rewarded path. Note that we have included new figures (see our responses below) that give a better example of what fast versus slower learning looks like. In addition, we now note in our Methods that we did not probe participants about their subjective awareness re: the perturbation:

“Note that, at the end of testing, we did not assess participants’ awareness of the manipulation (i.e., that they were, in fact, being rewarded based on a mirror image path of the visible path).”

Recommendation #2: Provide more behavioral quantification.(2a) The authors chose to only plot the average learning score in Figure 1D, without an indication of movement variability. I think this is quite important, to give the reader an impression of how variable the movements were at baseline, during early learning, and over the course of learning. There is evidence that baseline variability influences the 'detectability' of imposed rotations (in the case of adaptation learning), which could be relevant here. Shading the plots by movement variability would also be important to see if there was some refinement of the moment after participants performed at the ceiling (which seems to be the case ~ after trial 150). This is especially worrying given that in Fig 6A there is a clear indication that there is a large difference between subjects' solutions on the task. One subject exhibits almost a one-shot learning curve (reaching a score of 75 after one or two trials), whereas others don't seem to really learn until the near end. What does this between-subject variability mean for the authors' hypothesized neural processes?

In line with these recommendations, we have now provided much better behavioral quantification of subject-level performance in both the main manuscript and supplementary material. For instance, in a new supplemental Figure 1 (shown below), we now include mean subject (+/- SE) reaction times (RTs), movement times (MTs) and movement path variability (our computing of these measures are now defined in our Methods section).

As can be seen in the figure, all three of these variables tended to decrease over the course of the study, though we note there was a noticeable uptick in both RTs and MTs from the Baseline to Early learning phase, once subjects started receiving trial-by-trial reward feedback based on their movements. With respect to path variability, it is not obvious that there was a significant refinement of the paths created during late learning (panel D below), though there was certainly a general trend for path variability to decrease over learning.

**Author response image 3. sa4fig3:** Behavioral measures of learning across the task. (A-D) shows average participant reward scores (A), reaction times (B), movement times (C) and path variability (D) over the course of the task. In each plot, the black line denotes the mean across participants and the gray banding denotes +/- 1 SEM. The three equal-length task epochs for subsequent neural analyses are indicated by the gray shaded boxes.

In addition to these above results, we have also created a new Figure 6 in the main manuscript, which now solely focuses on individual differences in subject learning (see below). Hopefully, this figure clarifies key features of the task and its reward structure, and also depicts (in movement trajectory space) what fast versus slow learning looks like in the task. Specifically, we believe that this figure now clearly delineates for the reader the mapping between movement trajectory and the reward score feedback presented to participants, which appeared to be a source of confusion based on the reviewers’ comments below. As can be clearly observed in this figure, trajectories that approximated the ‘visible path’ (black line) resulted in fairly mediocre scores (see score color legend at right), whereas trajectories that approximated the ‘reward path’ (dashed black line, see trials 191-200 of the fast learner) resulted in fairly high scores. This figure also more clearly delineates how fPCA loadings derived from our functional data analysis were used to derive subject-level learning scores (panel C).

**Author response image 4. sa4fig4:** Individual differences in subject learning performance. (A) Examples of a good learner (bordered in green) and poor learner (bordered in red). (B) Individual subject learning curves for the task. Solid black line denotes the mean across all subjects whereas light gray lines denote individual participants. The green and red traces denote the learning curves for the example good and poor learners denoted in A. (C) Derivation of subject learning scores. We performed functional principal component analysis (fPCA) on subjects’ learning curves in order to identify the dominant patterns of variability during learning. The top component, which encodes overall learning, explained the majority of the observed variance (~75%). The green and red bands denote the effect of positive and negative component scores, respectively, relative to mean performance. Thus, subjects who learned more quickly than average have a higher loading (in green) on this ‘Learning score’ component than subjects who learned more slowly (in red) than average. The plot at right denotes the loading for each participant (open circles) onto this Learning score component.

The reviewers note that there are large individual differences in learning performance across the task. This was clearly our hope when designing the reward structure of this task, as it would allow us to further investigate the neural correlates of these individual differences (indeed, during pilot testing, we sought out a reward structure to the task that would allow for these intersubject differences). The subjects who learn early during the task end up having higher fPCA scores than the subjects who learn more gradually (or learn the task late). From our perspective, these differences are a feature, and not a bug, and they do not negate any of our original interpretations. That is, subjects who learn earlier on average tend to contract their DAN-A network during the early learning phase whereas subjects who learn more slowly on average (or learn late) instead tend to contract their DAN-A network during late learning (Fig. 7).

(2b) In the methods, the authors stated that they scaled the score such that even a perfectly traced visible path would always result in an imperfect score of 40 patients. What happens if a subject scores perfectly on the first try (which seemed to have happened for the green highlighted subject in Fig 6A), but is then permanently confronted with a score of 40 or below? Wouldn't this result in an error-clamp-like (error-based motor adaptation) design for this subject and all other high performers, which would vastly differ from the task demands for the other subjects? How did the authors factor in the wide between-subject variability?

We think the reviewers may have misinterpreted the reward structure of the task, and we apologize for not being clearer in our descriptions. The reward score that subjects received after each trial was based on how well they traced the mirror-image of the visible path. However, all the participant can see on the screen is the visible path. We hope that our inclusion of the new Figure 6 (shown above) makes the reward structure of the task, and its relationship to movement trajectories, much clearer. We should also note that, even for the highest performing subject (denoted in Fig. 6), it still required approximately 20 trials for them to reach asymptote performance.

(2c) The study would benefit from a more detailed description of participants' behavioral performance during the task. Specifically, it is crucial to understand how participants' motor skills evolve over time. Information on changes in movement speed, accuracy, and other relevant behavioral metrics would enhance the understanding of the relationship between behavior and brain activity during the learning process. Additionally, please clarify whether the display on the screen was presented continuously throughout the entire trial or only during active movement periods. Differences in display duration could potentially impact the observed differences in brain activity during learning.

We hope that with our inclusion of the new Supplementary Figure 1 (shown above) this addresses the reviewers’ recommendation. Generally, we find that RTs, MTs and path variability all decrease over the course of the task. We think this relates to the early learning phase being more attentionally demanding and requiring more conscious effort, than the later learning phases.

Also, yes, the visible path was displayed on the screen continuously throughout the trial, and only disappeared at the 4.5 second mark of each trial (when the screen was blanked and the data was saved off for 1.5 seconds prior to commencement of the next trial; 6 seconds total per trial). Thus, there were no differences in display duration across trials and phases of the task. We have now clarified this in the Methods section, where we now write the following:

“When the cursor reached the target distance, the target changed color from red to green to indicate that the trial was completed. Importantly, other than this color change in the distance marker, the visible curved path remained constant and participants never received any feedback about the position of their cursor.”

(2d) It is unclear from plots 6A, 6B, and 1D how the scale of the behavioral data matches with the scaling of the scores. Are these the 'real' scores, meaning 100 on the y-axis would be equivalent to 40 in the task? Why then do all subjects reach an asymptote at 75? Or is 75 equivalent to 40 and the axis labels are wrong?

As indicated above, we clearly did a poor job of describing the reward structure of our task in our original paper, and we now hope that our inclusion of Figure 6 makes things clear. A ‘40’ score on the y-axis would indicate that a subject has perfectly traced the visible path whereas a perfect ‘100’ score would indicate that a subject has perfectly traced the (hidden) mirror image path.

The fact that several of the subjects reach asymptote around 75 is likely a byproduct of two factors. Firstly, the subjects performed their movements in the absence of any visual error feedback (they could not see the position of a cursor that represented their hand position), which had the effect of increasing motor variability in their actions from trial to trial. Secondly, there appears to be an underestimation among subjects regarding the curvature of the concealed, mirror-image path (i.e., that the rewarded path actually had an equal but opposite curvature to that of the visible path). This is particularly evident in the case of the top-performing subject (illustrated in Figure 6A) who, even during late learning, failed to produce a completely arched movement.

(2e) Labeling of Contrasts: There is a consistent issue with the labeling of contrasts in the presented figures, causing confusion. While the text refers to the difference as "baseline to early learning," the label used in figures, such as Figure 4, reads "baseline > early." It is essential to clarify whether the presented contrast is indeed "baseline > early" or "early > baseline" to avoid any misinterpretation.

We thank the reviewers for catching this error. Indeed, the intended label was Early > Baseline, and this has now been corrected throughout.

Recommendation #3. Clarify which motor learning mechanism(s) are at play.(3a) Participants were performing at a relatively low level, achieving around 50-60 points by the end of learning. This outcome may not be that surprising, given that reward-based learning might have a substantial explicit component and may also heavily depend on reasoning processes, beyond reinforcement learning or contextual recall (Holland et al., 2018; Tsay et al., 2023). Even within our own data, where explicit processes are isolated, average performance is low and many individuals fail to learn (Brudner et al., 2016; Tsay et al., 2022). Given this, many participants in the current study may have simply given up. A potential indicator of giving up could be a subset of participants moving straight ahead in a rote manner (a heuristic to gain moderate points). Consequently, alterations in brain networks may not reflect exploration and exploitation strategies but instead indicate levels of engagement and disengagement. Could the authors plot the average trajectory and the average curvature changes throughout learning? Are individuals indeed defaulting to moving straight ahead in learning, corresponding to an average of 50-60 points? If so, the interpretation of brain activity may need to be tempered.

We can do one better, and actually give you a sense of the learning trajectories for every subject over time. In the figure below, which we now include as Supplementary Figure 2 in our revision, we have plotted, for each subject, a subset of their movement trajectories across learning trials (every 10 trials). As can be seen in the diversity of these trajectories, the average trajectory and average curvature would do a fairly poor job of describing the pattern of learning-related changes across subjects. Moreover, it is not obvious from looking at these plots the extent to which poor learning subjects (i.e., subjects who never converge on the reward path) actually ‘give up’ in the task — rather, many of these subjects still show some modulation (albeit minor) of their movement trajectories in the later trials (see the purple and pink traces). As an aside, we are also not entirely convinced that straight ahead movements, which we don’t find many of in our dataset, can be taken as direct evidence that the subject has given up.

**Author response image 5. sa4fig5:** Variability in learning across subjects. Plots show representative trajectory data from each subject (n=36) over the course of the 200 learning trials. Coloured traces show individual trials over time (each trace is separated by ten trials, e.g., trial 1, 10, 20, 30, etc.) to give a sense of the trajectory changes throughout the task (20 trials in total are shown for each subject).

We should also note that we are not entirely opposed to the idea of describing aspects of our findings in terms of subject engagement versus disengagement over time, as such processes are related at some level to exploration (i.e., cognitive engagement in finding the best solution) and exploitation (i.e., cognitively disengaging and automating one’s behavior). As noted in our reply to Recommendation #1 above, we now give some consideration of these explanations in our Discussion section, where we now write:

“It is also possible that these task-related shifts in connectivity relates to shifts in task-general processes, such as changes in the allocation of attentional resources (Bédard and Song, 2013; Rosenberg et al., 2016) or overall cognitive engagement (Aben et al., 2020), which themselves play critical roles in shaping learning (Codol et al., 2018; Holland et al., 2018; Song, 2019; Taylor and Thoroughman, 2008, 2007; for a review of these topics, see Tsay et al., 2023). Such processes are particularly important during the earlier phases of learning when sensorimotor contingencies need to be established. While these remain questions for future work, our data nevertheless suggest that this shift in connectivity may be enabled through the PMC.”

(3b) The authors are mixing two commonly used paradigms, reward-based learning, and motor adaptation, but provide no discussion of the different learning processes at play here. Which processes were they attempting to probe? Making this explicit would help the reader understand which brain regions should be implicated based on previous literature. As it stands, the task is hard to interpret. Relatedly, there is a wealth of literature on explicit vs implicit learning mechanisms in adaptation tasks now. Given that the authors are specifically looking at brain structures in the cerebral cortex that are commonly associated with explicit and strategic learning rather than implicit adaptation, how do the authors relate their findings to this literature? Are the learning processes probed in the task more explicit, more implicit, or is there a change in strategy usage over time? Did the authors acquire data on strategies used by the participants to solve the task? How does the baseline variability come into play here?

As noted in our paper, our task was directly inspired by the reward-based motor learning tasks developed by Dam et al., 2013 (Plos One) and Wu et al., 2014 (Nature Neuroscience). What drew us to these tasks is that they allowed us to study the neural bases of reward-based learning mechanisms in the absence of subjects also being able to exploit error-based mechanisms to achieve learning. Indeed, when first describing the task in the Results section of our paper we wrote the following:

“Importantly, because subjects received no visual feedback about their actual finger trajectory and could not see their own hand, they could only use the score feedback — and thus only reward-based learning mechanisms — to modify their movements from one trial to the next (Dam et al., 2013; Wu et al., 2014).”

If the reviewers are referring to ‘motor adaptation’ in the context in which that terminology is commonly used — i.e., the use of sensory prediction errors to support error-based learning — then we would argue that motor adaptation is not a feature of the current study. It is true that in our study subjects learn to ‘adapt’ their movements across trials, but this shaping of the movement trajectories must be supported through reinforcement learning mechanisms (and, of course, supplemented by the use of cognitive strategies as discussed in the nice review by Tsay et al., 2023). We apologize for not being clearer in our paper about this key distinction and we have now included new text in the introduction to our Results to directly address this:

“Importantly, because subjects received no visual feedback about their actual finger trajectory and could not see their own hand, they could only use the score feedback — and thus only reward-based learning mechanisms — to modify their movements from one trial to the next (Dam et al., 2013; Wu et al., 2014). That is, subjects could not use error-based learning mechanisms to achieve learning in our study, as this form of learning requires sensory errors that convey both the change in direction and magnitude needed to correct the movement.”

With this issue aside, we are well aware of the established framework for thinking about sensorimotor adaptation as being composed of a combination of explicit and implicit components (indeed, this has been a central feature of several of our other recent neuroimaging studies that have explored visuomotor rotation learning, e.g., Gale et al., 2022 PNAS, Areshenkoff et al., 2022 elife, Standage et al., 2023 Cerebral Cortex). However, there has been comparably little work done on these parallel components within the domain of reinforcement learning tasks (though see Codol et al., 2018; Holland et al., 2018, van Mastrigt et al., 2023; see also the Tsay et al., 2023 review), and as far as we can tell, nothing has been done to date in the reward-based motor learning area using fMRI. By design, we avoided using descriptors of ‘explicit’ or ‘implicit’ in our study because our experimental paradigm did not allow a separate measurement of those two components to learning during the task. Nevertheless, it seems clear to us from examining the subjects’ learning curves (see supplementary figure 2 above), that individuals who learn very quickly are using strategic processes (such as action exploration to identify the best path) to enhance their learning. As we noted in an above response, we did not query subjects after the fact about their strategy use, which admittedly was a missed opportunity on our part.

**Author response image 6. sa4fig6:** 

With respect to the comment on baseline variability and its relationship to performance, this is an interesting idea and one that was explored in the Wu et al., 2014 Nature Neuroscience paper. Prompted by the reviewers, we have now explored this idea in the current data set by testing for a relationship between movement path variability during baseline trials (all 70 baseline trials, see Supplementary Figure 1D above for reference) and subjects’ fPCA score on our learning task. However, when we performed this analysis, we did not observe a significant positive relationship between baseline variability and subject performance. Rather, we actually found a trend towards a negative relationship (though this was non-significant; r=-0.2916, p=0.0844). Admittedly, we are not sure what conclusions can be drawn from this analysis, and in any case, we believe it to be tangential to our main results. We provide the results (at right) for the reviewers if they are interested. This may be an interesting avenue for exploration in future work.

Recommendation #4: Provide stronger justification for brain imaging methods.(4a) Observing how brain activity varies across these different networks is remarkable, especially how sensorimotor regions separate and then contract with other, more cognitive areas. However, does the signal-to-noise ratio in each area/network influence manifold eccentricity and limit the possible changes in eccentricity during learning? Specifically, if a region has a low signal-to-noise ratio, it might exhibit minimal changes during learning (a phenomenon perhaps relevant to null manifold changes in the striatum due to low signal-to-noise); conversely, regions with higher signal-to-noise (e.g., motor cortex in this sensorimotor task) might exhibit changes more easily detected. As such, it is unclear how to interpret manifold changes without considering an area/network's signal-to-noise ratio.

We appreciate where these concerns are coming from. First, we should note that the timeseries data used in our analysis were z-transformed (mean zero, 1 std) to allow normalization of the signal both over time and across regions (and thus mitigate the possibility that the changes observed could simply reflect mean overall signal changes across different regions). Nevertheless, differences in signal intensity across brain regions — particularly between cortex and striatum — are well-known, though it is not obvious how these differences may manifest in terms of a task-based modulation of MR signals.

To examine this issue in the current data set, we extracted, for each subject and time epoch (Baseline, Early and Late learning) the raw scanner data (in MR arbitrary units, a.u.) for the cortical and striatal regions and computed the (1) mean signal intensity, (2) standard deviation of the signal (Std) and (3) temporal signal to noise ratio (tSNR; calculated by mean/Std). Note that in the fMRI connectivity literature tSNR is often the preferred SNR measure as it normalizes the mean signal based on the signal’s variability over time, thus providing a general measure of overall ‘signal quality’. The results of this analysis, averaged across subjects and regions, is shown below.

**Author response image 7. sa4fig7:** 

Note that, as expected, the overall signal intensity (left plot) of cortex is higher than in the striatum, reflecting the closer proximity of cortex to the receiver coils in the MR head coil. In fact, the signal intensity in cortex is approximately 38% higher than that in the striatum (~625 - 450/450). However, the signal variation in cortex is also greater than striatum (middle plot), but in this case approximately 100% greater (i.e., (~5 - 2.5)/2.5). The result of this is that the tSNR (mean/std) for our data set and the ROI parcellations we used is actually greater in the striatum than in cortex (right plot). Thus, all else being equal, there seems to have been sufficient tSNR in the striatum for us to have detected motor-learning related effects. As such, we suspect the null effects for the striatum in our study actually stem from two sources.

The first likely source is the relatively lower number of striatal regions (12) as compared to cortical regions (998) used in our analysis, coupled with our use of PCA on these data (which, by design, identifies the largest sources of variation in connectivity). In future studies, this unbalance could be rectified by using finer parcellations of the striatum (even down to the voxel level) while keeping the same parcellation of cortex (i.e., equate the number of ‘regions’ in each of striatum and cortex). The second likely source is our use of a striatal atlas (the Harvard-Oxford atlas) that divides brain regions based on their neuroanatomy rather than their function. In future work, we plan on addressing this latter concern by using finer, more functionally relevant parcellations of striatum (such as in Tian et al., 2020, Nature Neuroscience). Note that we sought to capture these interrelated possible explanations in our Discussion section, where we wrote the following:

“While we identified several changes in the cortical manifold that are associated with reward-based motor learning, it is noteworthy that we did not observe any significant changes in manifold eccentricity within the striatum. While clearly the evidence indicates that this region plays a key role in reward-guided behavior (Averbeck and O’Doherty, 2022; O’Doherty et al., 2017), there are several possible reasons why our manifold approach did not identify this collection of brain areas. First, the relatively small size of the striatum may mean that our analysis approach was too coarse to identify changes in the connectivity of this region. Though we used a 3T scanner and employed a widely-used parcellation scheme that divided the striatum into its constituent anatomical regions (e.g., hippocampus, caudate, etc.), both of these approaches may have obscured important differences in connectivity that exist within each of these regions. For example, areas such the hippocampus and caudate are not homogenous areas but themselves exhibit gradients of connectivity (e.g., head versus tail) that can only be revealed at the voxel level (Tian et al., 2020; Vos de Wael et al., 2021). Second, while our dimension reduction approach, by design, aims to identify gradients of functional connectivity that account for the largest amounts of variance, the limited number of striatal regions (as compared to cortex) necessitates that their contribution to the total whole-brain variance is relatively small. Consistent with this perspective, we found that the low-dimensional manifold architecture in cortex did not strongly depend on whether or not striatal regions were included in the analysis (see Supplementary Fig. 6). As such, selective changes in the patterns of functional connectivity at the level of the striatum may be obscured using our cortex x striatum dimension reduction approach. Future work can help address some of these limitations by using both finer parcellations of striatal cortex (perhaps even down to the voxel level)(Tian et al., 2020) and by focusing specifically on changes in the interactions between the striatum and cortex during learning. The latter can be accomplished by selectively performing dimension reduction on the slice of the functional connectivity matrix that corresponds to functional coupling between striatum and cortex.”

(4b) Could the authors clarify how activity in the dorsal attention network (DAN) changes throughout learning, and how these changes also relate to individual differences in learning performance? Specifically, on average, the DAN seems to expand early and contract late, relative to the baseline. This is interpreted to signify that the DAN exhibits lesser connectivity followed by greater connectivity with other brain regions. However, in terms of how these changes relate to behavior, participants who go against the average trend (DAN exhibits more contraction early in learning, and expansion from early to late) seem to exhibit better learning performance. This finding is quite puzzling. Does this mean that the average trend of expansion and contraction is not facilitative, but rather detrimental, to learning? [Another reviewer added: The authors do not state any explicit hypotheses, but only establish that DMN coordinates activity among several regions. What predictions can we derive from this? What are the authors looking for in the data? The work seems more descriptive than hypothesis-driven. This is fine but should be clarified in the introduction.]

These are good questions, and we are glad the reviewers appreciated the subtlety here. The reviewers are indeed correct that the relationship of the DAN-A network to behavioral performance appears to go against the grain of the group-level results that we found for the entire DAN network (which we note is composed of both the DAN-A and DAN-B networks). That is, subjects who exhibited greater contraction from Baseline to Early learning and likewise, greater expansion from Early to Late learning, tended to perform better in the task (according to our fPCA scores). However, on this point it is worth noting that it was mainly the DAN-B network which exhibited group-level expansion from Baseline to Early Learning whereas the DAN-A network exhibited negligible expansion. This can be seen in Author response image 8 below, which shows the pattern of expansion and contraction (as in Fig. 4), but instead broken down into the 17-network parcellation. The red asterisk denotes the expansion from Baseline to Early learning for the DAN-B network, which is much greater than that observed for the DAN-A network (which is basically around the zero difference line).

**Author response image 8. sa4fig8:** 

Thus, it appears that the DAN-A and DAN-B networks are modulated to a different extent during the task, which likely contributes to the perceived discrepancy between the group-level effects (reported using the 7-network parcellation) and the individual differences effects (reported using the finer 17-network parcellation). Based on the reviewers’ comments, this seems like an important distinction to clarify in the manuscript, and we have now described this nuance in our Results section where we now write:

“...Using this permutation testing approach, we found that it was only the change in eccentricity of the DAN-A network that correlated with Learning score (see Fig. 7C), such that the more the DAN-A network decreased in eccentricity from Baseline to Early learning (i.e., contracted along the manifold), the better subjects performed at the task (see Fig. 7C, scatterplot at right). Consistent with the notion that changes in the eccentricity of the DAN-A network are linked to learning performance, we also found the inverse pattern of effects during Late learning, whereby the more that this same network increased in eccentricity from Early to Late learning (i.e., expanded along the manifold), the better subjects performed at the task (Fig. 7D). We should note that this pattern of performance effects for the DAN-A — i.e., greater contraction during Early learning and greater expansion during Late learning being associated with better learning — appears at odds with the group-level effects described in Fig. 4A and B, where we generally find the opposite pattern for the entire DAN network (composed of the DAN-A and DAN-B subnetworks). However, this potential discrepancy can be explained when examining the changes in eccentricity using the 17-network parcellation (see Supplementary Figure 8). At this higher resolution level we find that these group-level effects for the entire DAN network are being largely driven by eccentricity changes in the DAN-B network (areas in anterior superior parietal cortex and premotor cortex), and not by mean changes in the DAN-A network. By contrast, our present results suggest that it is the contraction and expansion of areas of the DAN-A network (and not DAN-B network) that are selectively associated with differences in subject learning performance.”

Finally, re: the reviewers’ comments that we do not state any explicit hypotheses etc., we acknowledge that, beyond our general hypothesis stated at the outset about the DMN being involved in reward-based motor learning, our study is quite descriptive and exploratory in nature. Such little work has been done in this research area (i.e., using manifold learning approaches to study motor learning with fMRI) that it would be disingenuous to have any stronger hypotheses than those stated in our Introduction. Thus, to make the exploratory nature of our study clear to the reader, we have added the following text (in red) to our Introduction:

“Here we applied this manifold approach to explore how brain activity across widely distributed cortical and striatal systems is coordinated during reward-based motor learning. We were particularly interested in characterizing how connectivity between regions within the DMN and the rest of the brain changes as participants shift from learning the relationship between motor commands and reward feedback, during early learning, to subsequently using this information, during late learning. We were also interested in exploring whether learning-dependent changes in manifold structure relate to variation in subject motor performance.”

We hope these changes now make it obvious the intention of our study.

(4c) The paper examines a type of motor adaptation task with a reward-based learning component. This, to me, strongly implicates the cerebellum, given that it has a long-established crucial role in adaptation and has recently been implicated in reward-based learning (see work by Wagner & Galea). Why is there no mention of the cerebellum and why it was left out of this study? Especially given that the authors state in the abstract they examine cortical and subcortical structures. It's evident from the methods that the authors did not acquire data from the cerebellum or had too small a FOV to fully cover it (34 slices at 4 mm thickness 136 mm which is likely a bit short to fully cover the cerebellum in many participants). What was the rationale behind this methodological choice? It would be good to clarify this for the reader. Related to this, the authors need to rephrase their statements on 'whole-brain' connectivity matrices or analyses - it is not whole-brain when it excludes the cerebellum.

As we noted above, we do not believe this task to be a motor adaptation task, in the sense that subjects are not able to use sensory prediction errors (and thus error-based learning mechanisms) to improve their performance. Rather, by denying subjects this sensory error feedback they are only able to use reinforcement learning processes, along with cognitive strategies (nicely covered in Tsay et al., 2023), to improve performance. Nevertheless, we recognize that the cerebellum has been increasingly implicated in facets of reward-based learning, particularly within the rodent domain (e.g., Wagner et al., 2017; Heffley et al., 2018; Kostadinov et al., 2019, etc.). In our study, we did indeed collect data from the cerebellum but did not include it in our original analyses, as we wanted (1) the current paper to build on prior work in the human and macaque reward-learning domain (which focuses solely on striatum and cortex, and which rarely discusses cerebellum, see Averbeck & O’Doherty, 2022 & Klein-Flugge et al., 2022 for recent reviews), and, (2) allow this to be a more targeted focus of future work (specifically we plan on focusing on striatal-cerebellar interactions during learning, which are hypothesized based on the neuroanatomical tract tracing work of Bostan and Strick, etc.). We hope the reviewers respect our decisions in this regard.

Nevertheless, we acknowledge that based on our statements about ‘whole-brain’ connectivity and vagueness about what we mean by ‘subcortex,’ that this may be confusing for the reader. We have now removed and/or corrected such references throughout the paper (however, note that in some cases it is difficult to avoid reference to “whole-brain” — e.g., “whole-brain correlation map” or “whole-brain false discovery rate correction”, which is standard terminology in the field).

In addition, we are now explicit in our Methods section that the cerebellum was not included in our analyses.

“Each volume comprised 34 contiguous (no gap) oblique slices acquired at a ~30° caudal tilt with respect to the plane of the anterior and posterior commissure (AC-PC), providing whole-brain coverage of the cerebrum and cerebellum. Note that for the current study, we did not examine changes in cerebellar activity during learning.”

(4d) The authors centered the matrices before further analyses to remove variance associated with the subject. Why not run a PCA on the connectivity matrices and remove the PC that is associated with subject variance? What is the advantage of first centering the connectivity matrices? Is this standard practice in the field?

Centering in some form has become reasonably common in the functional connectivity literature, as there is considerable evidence that task-related (or cognitive) changes in whole-brain connectivity are dwarfed by static, subject-level differences (e.g., Gratton, et al, 2018, Neuron). If covariance matrices were ordinary scalar values, then isolating task-related changes could be accomplished simply by subtracting a baseline scan or mean score; but because the space of covariance matrices is non-Euclidean, the actual computations involved in this subtraction are more complex (see our Methods). However, fundamentally (and conceptually) our procedure is simply ordinary mean-centering, but adapted to this non-Euclidean space. Despite the added complexity, there is considerable evidence that such computations — adapted directly to the geometry of the space of covariance matrices — outperform simpler methods, which treat covariance matrices as arrays of real numbers (e.g. naive substraction, see Dodero et al. & Ng et al., references below). Moreover, our previous work has found that this procedure works quite well to isolate changes associated with different task conditions (Areshenkoff et al., 2021, Neuroimage; Areshenkoff et al., 2022, elife).

Although PCA can be adapted to work well with covariance matrix valued data, it would at best be a less direct solution than simply subtracting subjects' mean connectivity. This is because the top components from applying PCA would be dominated by both subject-specific effects (not of interest here), and by the large-scale connectivity structure typically observed in component based analyses of whole-brain connectivity (i.e. the principal gradient), whereas changes associated with task-condition (the thing of interest here) would be buried among the less reliable components. By contrast, our procedure directly isolates these task changes.

References cited above:

Dodero, L., Minh, H. Q., San Biagio, M., Murino, V., & Sona, D. (2015, April). Kernel-based classification for brain connectivity graphs on the Riemannian manifold of positive definite matrices. In 2015 IEEE 12th international symposium on biomedical imaging (ISBI) (pp. 42-45). IEEE.

Ng, B., Dressler, M., Varoquaux, G., Poline, J. B., Greicius, M., & Thirion, B. (2014). Transport on Riemannian manifold for functional connectivity-based classification. In Medical Image Computing and Computer-Assisted Intervention–MICCAI 2014: 17th International Conference, Boston, MA, USA, September 14-18, 2014, Proceedings, Part II 17 (pp. 405-412). Springer International Publishing.

(4e) Seems like a missed opportunity that the authors just use a single, PCA-derived measure to quantify learning, where multiple measures could have been of interest, especially given that the introduction established some interesting learning-related concepts related to exploration and exploitation, which could be conceptualized as movement variability and movement accuracy. It is unclear why the authors designed a task that was this novel and interesting, drawing on several psychological concepts, but then chose to ignore these concepts in the analysis.

We were disappointed to hear that the reviewers did not appreciate our functional PCA-derived measure to quantify subject learning. This is a novel data-driven analysis approach that we have previously used with success in recent work (e.g., Areshenkoff et al., 2022, elife) and, from our perspective, we thought it was quite elegant that we were able to describe the entire trajectory of learning across all participants along a single axis that explained the majority (~75%) of the variance in the patterns of behavioral learning data. Moreover, the creation of a single behavioral measure per participant (what we call a ‘Learning score’, see Fig. 6C) helped simplify our brain-behavior correlation analyses considerably, as it provided a single measure that accounts for the natural auto-correlation in subjects’ learning curves (i.e., that subjects who learn quickly also tend to be better overall learners by the end of the learning phase). It also avoids the difficulty (and sometimes arbitrariness) of having to select specific trial bins for behavioral analysis (e.g., choosing the first 5, 10, 20 or 25 trials as a measure of ‘early learning’, and so on). Of course, one of the major alternatives to our approach would have involved fitting an exponential to each subject’s learning curves and taking measures like learning rate etc., but in our experience we have found that these types of models don’t always fit well, or derive robust/reliable parameters at the individual subject level. To strengthen the motivation for our approach, we have now included the following text in our Results:

“To quantify this variation in subject performance in a manner that accounted the auto-correlation in learning performance over time (i.e., subjects who learned more quickly tend to exhibit better performance by the end of learning), we opted for a pure data-driven approach and performed functional principal component analysis (fPCA; (Shang, 2014)) on subjects’ learning curves. This approach allowed us to isolate the dominant patterns of variability in subject’s learning curves over time (see Methods for further details; see also Areshenkoff et al., 2022).”

In any case, the reviewers may be pleased to hear that in current work in the lab we are using more model-based approaches to attempt to derive sets of parameters (per participant) that relate to some of the variables of interest described by the reviewers, but that we relate to much more dynamical (shorter-term) changes in brain activity.

(4f) Overall Changes in Activity: The manuscript should delve into the potential influence of overall changes in brain activity on the results. The choice of using Euclidean distance as a metric for quantifying changes in connectivity is sensitive to scaling in overall activity. Therefore, it is crucial to discuss whether activity in task-relevant areas increases from baseline to early learning and decreases from early to late learning, or if other patterns emerge. A comprehensive analysis of overall activity changes will provide a more complete understanding of the findings.

These are good questions and we are happy to explore this in the data. However, as mentioned in our response to query 4a above, it is important to note that the timeseries data for each brain region was z-scored prior to analysis, with the aim of removing any mean changes in activity levels (note that this is a standard preprocessing step when performing functional connectivity analysis, given that mean signal changes are not the focus of interest in functional connectivity analyses).

To further emphasize these points, we have taken our z-scored timeseries data and calculated the mean signal for each region within each task epoch (Baseline, Early and Late learning, see panel A in figure below). The point of showing this data (where each z-score map looks near identical across the top, middle and bottom plots) is to demonstrate just how miniscule the mean signal changes are in the z-scored timeseries data. This point can also be observed when plotting the mean z-score signal across regions for each epoch (see panel B in figure below). Here we find that Baseline and Early learning have a near identical mean activation level across regions (albeit with slightly different variability across subjects), whereas there is a slight increase during late learning — though it should be noted that our y-axis, which measures in the thousandths, really magnifies this effect.

To more directly address the reviewers’ comments, using the z-score signal per region we have also performed the same statistical pairwise comparisons (Early > Baseline and Late>Early) as we performed in the main manuscript Fig. 4 (see panel C in Author response image 9 below). In this plot, areas in red denote an increase in activity from Baseline to Early learning (top plot) and from Early to Late learning (bottom plot), whereas areas in blue denote a decrease for those same comparisons. The important thing to emphasize here is that the spatial maps resulting from this analysis are generally quite different from the maps of eccentricity that we report in Fig. 4 in our paper. For instance, in the figure below, we see significant changes in the activity of visual cortex between epochs but this is not found in our eccentricity results (compare with Fig. 4). Likewise, in our eccentricity results (Fig. 4), we find significant changes in the manifold positioning of areas in medial prefrontal cortex (MPFC), but this is not observed in the activation levels of these regions (panel C below). Again, we are hesitant to make too much of these results, as the activation differences denoted as significant in the figure below are likely to be an effect on the order of thousandths of a z-score (e.g., 0.002 > 0.001), but this hopefully assuages reviewers’ concerns that our manifold results are solely attributable to changes in overall activity levels.

We are hesitant to include the results below in our paper as we feel that they don’t add much to the interpretation (as the purpose of z-scoring was to remove large activation differences). However, if the reviewers strongly believe otherwise, we would consider including them in the supplement.

**Author response image 9. sa4fig9:** Examination of overall changes in activity across regions. (A) Mean z-score maps across subjects for the Baseline (top), Early Learning (middle) and Late learning (bottom) epochs. (B) Mean z-score across brain regions for each epoch. Error bars represent +/- 1 SEM. (C) Pairwise contrasts of the z-score signal between task epochs. Positive (red) and negative (blue) values show significant increases and decreases in z-score signal, respectively, following FDR correction for region-wise paired t-tests (at q<0.05).